# Exploring factors shaping antibiotic resistance patterns in *Streptococcus pneumoniae* during the 2020 COVID-19 pandemic

Aleksandra Kovacevic[1,2]\*, David RM Smith[1,2,3,4], Eve Rahbé[1,2], Sophie Novelli[2], Paul Henriot[3,5], Emmanuelle Varon[6], Robert Cohen[7,8,9,10,11], Corinne Levy[7,8,10,11], Laura Temime[3,5], Lulla Opatowski[1,2]

[1]Institut Pasteur, Université Paris Cité, Epidemiology and Modelling of Antibiotic Evasion (EMAE) unit, Paris, France; [2]Université Paris-Saclay, Université de Versailles Saint-Quentin-en-Yvelines, Inserm U1018, CESP, Anti-infective evasion and pharmacoepidemiology team, Montigny-Le-Bretonneux, France; [3]Modélisation, épidémiologie et surveillance des risques sanitaires (MESuRS), Conservatoire national des arts et métiers, Paris, France; [4]Health Economics Research Centre, Nuffield Department of Health, University of Oxford, Oxford, United Kingdom; [5]PACRI unit, Institut Pasteur, Conservatoire national des arts et métiers, Paris, France; [6]Centre National de Référence des Pneumocoques, Centre Hospitalier Intercommunal de Créteil, Créteil, France; [7]Institut Mondor de Recherche Biomédicale-Groupe de Recherche Clinique Groupe d'Etude des Maladies Infectieuses Néonatales et Infantiles (IMRB-GRC GEMINI), Université Paris Est, 94000, Créteil, France; [8]Groupe de Pathologie Infectieuse Pédiatrique (GPIP), 06200, Nice, France; [9]Unité Court Séjour, Petits Nourrissons, Service de Néonatologie, Centre Hospitalier, Intercommunal de Créteil, Créteil, France; [10]Association Clinique et Thérapeutique Infantile du Val-de-Marne (ACTIV), 94000, Créteil, France; [11]Association Française de Pédiatrie Ambulatoire (AFPA), 45000, Orléans, France

\*For correspondence: aleksandra.kovacevic@pasteur.fr

**Abstract** Non-pharmaceutical interventions implemented to block SARS-CoV-2 transmission in early 2020 led to global reductions in the incidence of invasive pneumococcal disease (IPD). By contrast, most European countries reported an increase in antibiotic resistance among invasive *Streptococcus pneumoniae* isolates from 2019 to 2020, while an increasing number of studies reported stable pneumococcal carriage prevalence over the same period. To disentangle the impacts of the COVID-19 pandemic on pneumococcal epidemiology in the community setting, we propose a mathematical model formalizing simultaneous transmission of SARS-CoV-2 and antibiotic-sensitive and -resistant strains of *S. pneumoniae*. To test hypotheses underlying these trends five mechanisms were built into the model and examined: (1) a population-wide reduction of antibiotic prescriptions in the community, (2) lockdown effect on pneumococcal transmission, (3) a reduced risk of developing an IPD due to the absence of common respiratory viruses, (4) community azithromycin use in COVID-19 infected individuals, (5) and a longer carriage duration of antibiotic-resistant pneumococcal strains. Among 31 possible pandemic scenarios involving mechanisms individually or in combination, model simulations surprisingly identified only two scenarios that reproduced the reported trends in the general population. They included factors (1), (3), and (4). These scenarios replicated a nearly 50% reduction in annual IPD, and an increase in antibiotic resistance from 20% to 22%, all while maintaining a relatively stable pneumococcal carriage. Exploring further, higher

SARS-CoV-2 $R_0$ values and synergistic within-host virus-bacteria interaction mechanisms could have additionally contributed to the observed antibiotic resistance increase. Our work demonstrates the utility of the mathematical modeling approach in unraveling the complex effects of the COVID-19 pandemic responses on AMR dynamics.

## Editor's evaluation

The mathematical modeling approach disentangles the impacts of the COVID-19 pandemic on antibiotic resistance in Streptococcus pneumoniae in the community setting and identifies the three most plausible driving mechanisms responsible for the observed trends in invasive isolates and pneumococcal carriage. The significance of the findings is important for our understanding of the changed epidemiology of invasive pneumococcal infections during the COVID19 pandemic. The strength of the evidence is graded as solid, derived from a well-described mathematical model evaluating multiple scenarios.

## Introduction

In the early 2020, international responses to the coronavirus disease 2019 (COVID-19) pandemic led to unprecedented worldwide change in population mixing, healthcare-seeking behavior, and infection prevention and control practices. This modified the ecology and epidemiology of many infectious diseases at a global scale. Strong impacts of COVID-19 on infectious disease dynamics have been reported for common viral and bacterial respiratory infections, sexually transmitted pathogens like HIV, vector-borne diseases like dengue, and even non-communicable diseases (*Braunstein et al., 2020*; *Brueggemann et al., 2021*; *Chen et al., 2022*; *Palmer et al., 2020*). Antimicrobial resistance (AMR), however, remains one of the leading threats to global health. In 2019, estimates showed that AMR in clinically relevant bacteria was associated with 4.95 million deaths, of which 1.27 million were described as directly attributable to resistance (*Murray et al., 2022*). Impacts of the COVID-19 pandemic on AMR dynamics remain relatively poorly understood.

A joint report from the WHO and European Centre for Disease Prevention and Control (ECDC) has reported 2020 AMR trends across 29 European countries for eight antibiotic-resistant bacterial pathogens of concern, including *S. pneumoniae* (*European Centre for Disease Prevention and Control and World Health Organization, 2022*). While the situation varies widely across bacterial species, antimicrobial groups, and regions, most European countries, including France, documented an increase in pneumococcal resistance to both penicillin and macrolides between 2019 and 2020. The resistance rates rose from 12.2% in 2019 to 15.6% in 2020 for penicillin and from 14.5% in 2019 to 16.9% in 2020 for macrolides, as reported in the EU/EEA (*European Centre for Disease Prevention and Control and World Health Organization, 2022*). However, increased pneumococcal resistance was accompanied by a sharp worldwide decline in invasive pneumococcal disease (IPD) incidence (*Brueggemann et al., 2021*; *Shaw et al., 2023*).

Similar declines in bacterial disease during early waves of COVID-19 have been observed in the context of sentinel community-acquired infections in New Zealand (*Duffy et al., 2021*), IPDs in Taiwan (*Chien et al., 2021*) and Hong Kong (*Teng et al., 2022*), and lower respiratory tract infections in China (*Chen et al., 2021*). Yet, surprisingly, a growing number of studies have reported mostly stable pneumococcal carriage throughout the COVID-19 pandemic containment, including among infants in Belgium (*Willen et al., 2021*), children in Vietnam (*Nation et al., 2023*), Serbia (*Petrović et al., 2022*), France (*Rybak et al., 2022*), South Africa (*Olwagen et al., 2024*), and Israel (*Dagan et al., 2023*), adults in Connecticut (*Wyllie et al., 2023*), and households in Seattle (*Bennett et al., 2023*). In contrast, a study conducted in Denmark reported a decrease in pneumococcal carriage among older adults during the COVID-19 lockdown (*Tinggaard et al., 2023*).

Understanding the cause of these trends is not straightforward, as many responses to the COVID-19 pandemic, such as the implementation of non-pharmaceutical interventions (NPIs), changes in healthcare-seeking behavior, and alterations in antibiotic prescribing, occurred over the period (*Knight et al., 2021*). To gain a comprehensive understanding of the changes in AMR epidemiology during the COVID-19 pandemic, it is essential to simultaneously consider a range of scales and indicators. These include the rates of incidence of invasive bacterial diseases (IBDs), the proportion of

antibiotic-resistant isolates among total invasive bacterial isolates, and the prevalence of asymptomatic bacterial carriage in healthy individuals.

Several mechanisms may underlie the explanation of these contrasting observations. First, NPIs implemented to block SARS-CoV-2 transmission, such as lockdowns and mask mandates, may have led to reduced bacterial transmission. Containment measures also massively reduced circulation of common respiratory viruses, which are known to be associated with IBD (*Domenech de Cellès et al., 2019*; *Smith and Opatowski, 2021*). Second, the lockdown was associated with reductions in primary care consultations (*Homeniuk and Collins, 2021*; *Read et al., 2023*; *Zhang et al., 2021*) leading to a global decrease of antibiotic prescriptions (*Högberg et al., 2021*). In contrast, frequent antibiotic prescribing to COVID-19 outpatients may have exacerbated AMR (*Clancy et al., 2020*; *Knight et al., 2021*). Differences in the duration of pneumococcal carriage may have also played a role (*Lehtinen et al., 2017*). Finally, potential within-host interactions between SARS-CoV-2 and *S. pneumoniae* could also have an impact on infection risk (*Amin-Chowdhury et al., 2021*), although strong evidence for such interactions remains limited (*Wong et al., 2023*).

Mathematical models incorporating the co-transmission of multiple pathogens within the same host population provide a framework for investigating different hypotheses that underlie the observed patterns in antibiotic resistance and incidence of IPD in *S. pneumoniae* and help to enhance our understanding of the mechanisms involved. Co-circulation models have been used previously to disentangle the public health consequences of interactions between pathogens such as influenza and *S. pneumoniae* (*Arduin et al., 2017*; *Domenech de Cellès et al., 2019*; *Shrestha et al., 2013*) and could similarly be used to understand impacts of the COVID-19 pandemic on pathogens coinciding with SARS-CoV-2. However, in a systematic PubMed search conducted on 4 December 2023, we identified no epidemiological models describing the simultaneous transmission of SARS-CoV-2 and antibiotic-resistant bacteria specific to the community setting (Appendix 1).

Here, to disentangle how the COVID-19 pandemic has impacted the epidemiological dynamics of antibiotic resistance in *S. pneumoniae*, we propose a mathematical model that formalizes the transmission of SARS-CoV-2 and both antibiotic-sensitive and -resistant strains of *S. pneumoniae* in the community setting, and which includes mechanistic impacts of COVID-19 burden on epidemiological parameters. Through simulation, we assess all possible combinations of these mechanisms to evaluate their overall impact on IPD incidence, antibiotic resistance, and the prevalence of pneumococcal carriage. Furthermore, we assess the changes in the incidence of antibiotic-resistant IPD as we vary the basic reproduction number ($R_0$) of SARS-CoV-2 during the first COVID-19 outbreak in Europe. We also consider assumed within-host pathogen interactions between SARS-CoV-2 and *S. pneumoniae*.

## Results

### Antibiotic resistance trends and incidence of invasive pneumococcal disease in 2020

In routine surveillance data reported to the European Antimicrobial Resistance Surveillance Network (EARS-Net), most European countries reported an increase in antibiotic resistance in *S. pneumoniae* from 2019 to 2020, as indicated by increases in the proportion of invasive isolates with phenotypic resistance to both penicillin and macrolides (*Figure 1A*). On the contrary, the total number of reported invasive isolates in the EU/EEA decreased by 44.3% from 2019 to 2020 (*European Centre for Disease Prevention and Control and World Health Organization, 2022*) suggesting a decrease in incidence of IPD (*Appendix 2—table 1*).

Invasive pneumococcal isolate data for France provided by the French National Reference Center for Pneumococci (CNRP) revealed similar trends. In France, the total number of reported invasive pneumococcal isolates decreased by 45.1% from 2019 to 2020 (from 1119–614), while antibiotic resistance in *S. pneumoniae* isolates to penicillin and macrolides showed an increasing trend from 26.2% in 2019 to 35.5% in 2020 for penicillin, and from 20.9% in 2019 to 23.0% in 2020 for macrolides (*Figure 1B*). General decreasing trend in antibiotic resistance from 2017 to 2019 in *S. pneumoniae* was interrupted in 2020 (*Figure 1—figure supplement 1*). These variations in antibiotic resistance manifested differently across age, with some age groups showing an increase in antibiotic resistance in 2020 compared to 2019, while others showed no significant change (*Figure 1B*).

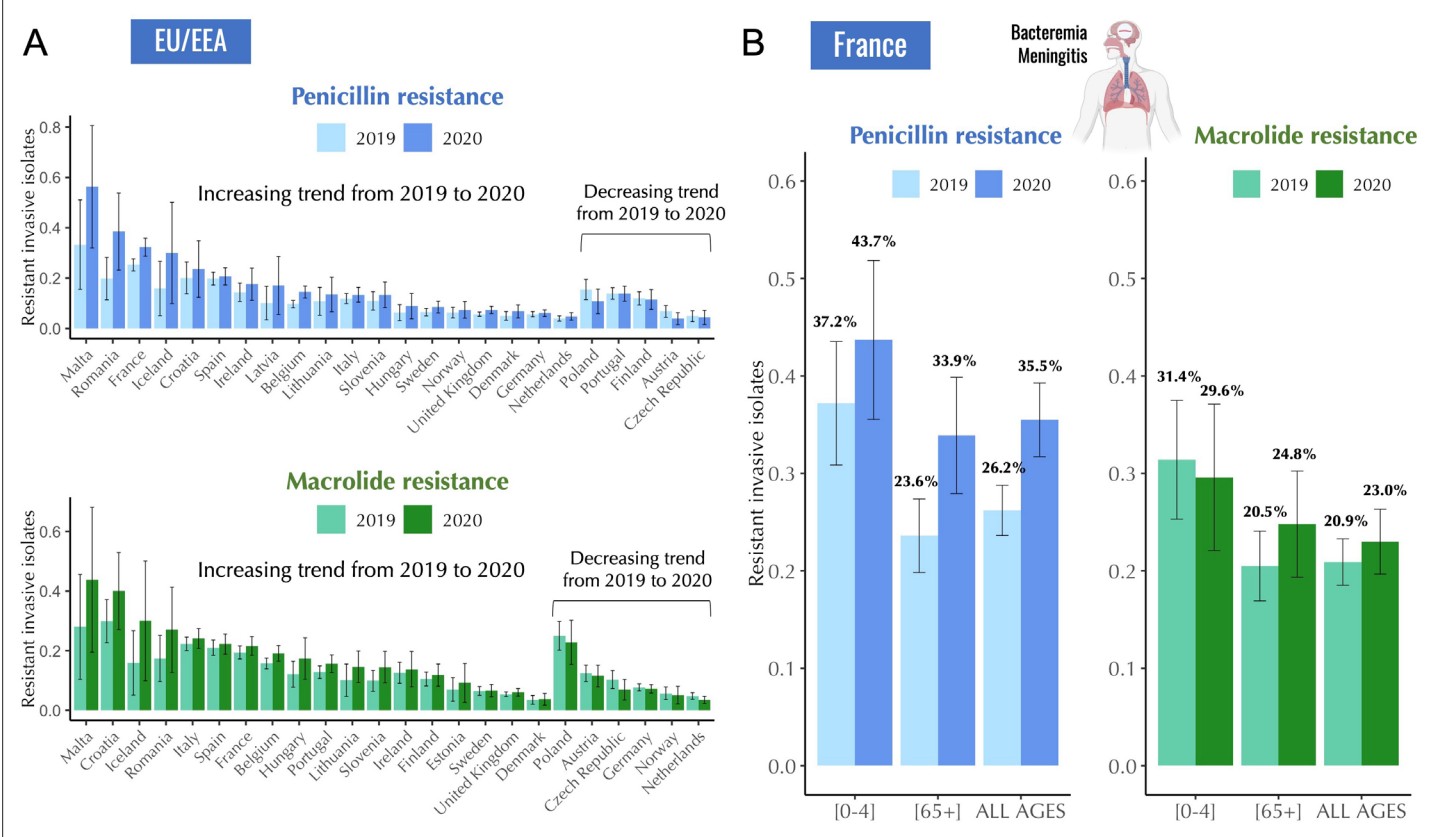

**Figure 1.** Antibiotic resistance trends in invasive *Streptococcus pneumoniae* isolates for the years 2019 and 2020. (**A**) The proportion of invasive *S. pneumoniae* isolates resistant to penicillin and macrolides (azithromycin/ clarithromycin/ erythromycin) reported to EARS-Net (European Antimicrobial Resistance Surveillance Network) for 24 European countries. Error bars show 95% confidence intervals. (**B**) The proportion of invasive *S. pneumoniae* isolates resistant to penicillin (MIC >0.064 mg/L) and macrolides (erythromycin) according to age. Error bars show 95% confidence intervals. The total number of invasive pneumococcal isolates reported in France decreased by 45.1% from 2019 to 2020 (from 1119–614). Data are provided by the French National Reference Center for Pneumococci.

The online version of this article includes the following source data and figure supplement(s) for figure 1:

**Source data 1.** European trends in antibiotic resistance in Streptococcus pneumoniae invasive isolates (2019-2020).

**Figure supplement 1.** Antibiotic resistance trends in invasive *Streptococcus pneumoniae* isolates in France, 2017–2020.

## Coinfection model of SARS-CoV-2 and *Streptococcus pneumoniae*

As mentioned above, several mechanisms may underlie the explanation of these contrasting observations (*Figure 2A*). COVID-19 NPIs may have led to reduced person-to-person bacterial transmission, potentially contributing to reduced rates of IPD incidence. These containment measures also massively reduced circulation of common respiratory viruses and the incidence of influenza-like-illnesses (ILIs). Respiratory viruses are known triggers and risk factors for developing an IBD from otherwise asymptomatic carriage; in that context, their reduction may have led to reduced infection risk (*Domenech de Cellès et al., 2019*; *Smith and Opatowski, 2021*). Due to reductions in primary care consultations in 2020, 26 European countries reported an estimated average decrease of 18.3% in overall antibiotic consumption, aligning with the global trend of reduced antibiotic prescriptions compared to 2019 (*Högberg et al., 2021*). On the other hand, frequent prescribing of azithromycin, a macrolide antibiotic initially hypothesized to be effective in COVID-19 treatment, has raised concerns for pandemic-associated antimicrobial overuse or misuse and may have exacerbated AMR during and following the first wave of the pandemic (*Clancy et al., 2020*; *Knight et al., 2021*; *Kournoutou and Dinos, 2022*; *Langford et al., 2021*; *PRINCIPLE Trial Collaborative Group, 2021*; *Rusic et al., 2021*). There are still uncertainties about pneumococcal ecology and the evolutionary processes that enable the robust coexistence of strains sensitive and resistant to antibiotics. The role of carriage duration, along with the impact of antibiotic consumption, is also not fully understood in this context. Longer carriage

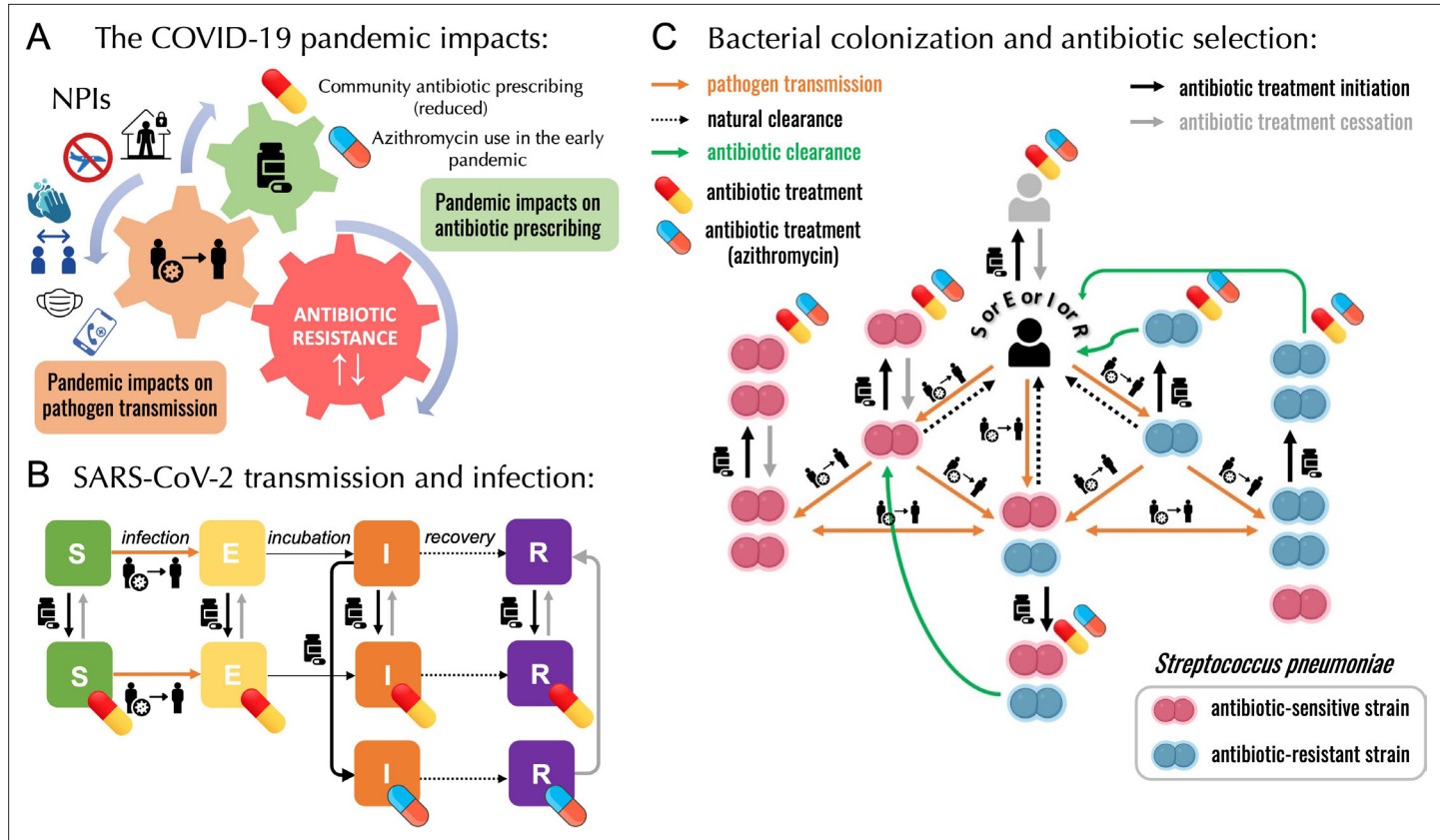

**Figure 2.** A modelling framework describing the transmission of SARS-CoV-2 and *Streptococcus pneumoniae* in the community setting, in the context of both general antibiotic prescribing and azithromycin prescribing for COVID-19 infected individuals. (**A**) Non-pharmaceutical interventions (NPIs) implemented to control SARS-CoV-2 transmission (lockdown, face mask use, improved hygiene practices, travel restrictions, quarantine, telemedicine, and physical distancing) may also modify transmission of other pathogens, in addition to impacting antibiotic prescribing due to altered inter-individual contact and health-care seeking behavior. (**B**) SEIR (Susceptible-Exposed-Infected-Recovered) model with antibiotic treatment compartments depicts interaction between SARS-CoV-2 infection and antibiotic prescribing, including both general community prescribing and azithromycin prescribing among individuals infected with SARS-CoV-2. (**C**) Diagram depicting how pneumococcal colonization and the community antibiotic prescribing are affected by the COVID-19 pandemic impacts. Initiation of antibiotic treatment is assumed independent of bacterial carriage, reflecting widespread bystander selection for commensal bacteria like *S. pneumoniae*. For a complete modeling framework, see section S2 in Supporting Information.

duration of antibiotic-resistant pneumococcal strains is a proposed explanation for this coexistence (*Lehtinen et al., 2017*). If so, antibiotic-resistant pneumococcal strains may have had an advantage during the lockdown period due to smaller clearance rates, ultimately leading to an increase in antibiotic resistance. Finally, among individuals with COVID-19, potential within-host interactions between SARS-CoV-2 and *S. pneumoniae* could also have had an impact on bacterial colonization and infection dynamics (*Amin-Chowdhury et al., 2021*).

To test mechanistic impacts of responses to the COVID-19 pandemic on pneumococcal epidemiology, we developed a compartmental, deterministic transmission model describing infection with SARS-CoV-2 being introduced on 1 Jan 2020 (*Figure 2B*) after colonization with *S. pneumoniae* reached an equilibrium in a large, well-mixed human population (*Figure 2C*). Two lockdowns were implemented in the model in agreement with the two lockdowns implemented in France in 2020. The model was parameterized to *S. pneumoniae* and five mechanisms were built in into the model: (1) a population-wide reduction of antibiotic prescriptions in the community by 18% due to the reduced healthcare-seeking behavior, (2) lockdown reducing pneumococcal transmission by 25%, (3) a reduced risk of developing an IPD from asymptomatic carriage due to the absence of common respiratory viruses during the first lockdown (reduced by a factor $IPD_{risk} = 0.2$), which continues after the first lockdown, albeit at a diminished level ($IPD_{risk} = 0.4$), (4) community azithromycin use in 10% of COVID-19 infected individuals, (5) and a longer carriage duration of antibiotic-resistant pneumococcal strains giving them a fitness advantage over antibiotic-resistant strains (40 vs 30 days).

## Exploring the mechanisms and identifying the optimal scenario for explaining reported trends

We conducted assessments on five distinct hypotheses, each characterized by a precise underlying mechanism, and explored these hypotheses in combination within 31 pandemic scenarios, along with two pre-pandemic (baseline) scenarios, which assume no SARS-CoV-2 circulation in the population and allow for the same 30-day carriage duration (pre-pandemic 1) of both antibiotic-sensitive and -resistant strains ($d_s = d_R$) or a longer, 40-day carriage duration (pre-pandemic 2) of -resistant strains ($d_s > d_R$) (**Table 1**).

We assessed how different combinations of mechanisms may impact: (i) a change in the annual IPD incidence as compared to the pre-pandemic (baseline) period, (ii) antibiotic resistance rate in IPDs, defined as the annual number of antibiotic-resistant IPD cases over the total number of IPD cases, and (iii) daily prevalences of antibiotic-resistant and total pneumococcal carriage in a simulated population of 100,000 individuals (see **Appendix 2—table 2** for parameter values). To identify scenarios most compatible with the reported trends, results from model simulations were compared to reported data trends from France in 2020 and more broadly to general EU/EEA reported trends that followed similar patterns. Surprisingly only two scenarios were compatible with reported trends. Scenarios S19 and S29 univocally reproduced increased antibiotic resistance in the general population (AR%) accompanied by a reduction in the annual IPD incidence by almost 50% (IPD inc) with generally stable pneumococcal carriage prevalence in healthy individuals during lockdown (Sp.). In contrast, model simulations revealed that a reduction in the community antibiotic consumption alone (–18%) could not explain the reported trends and generally led to a reduction of antibiotic resistance (**Table 1**, see scenario S1). Assuming a longer duration of antibiotic-resistant pneumococcal carriage alone did not explain either the rise in antibiotic resistance (**Table 1**, see scenario S5). Hypothesizing that lockdown reduced the transmission of pneumococcal carriage (by 25%) in addition to a reduced community antibiotic prescribing did not seem probable since, in simulations, this yielded a major reduction in pneumococcal carriage during containment measures in all scenarios where this mechanism was implemented. On the other hand, considering an indirect impact of lockdown on pneumococcal carriage where we implemented a reduction factor for the risk of developing and IPD from otherwise asymptomatic carriage due to the absence of viral respiratory infections during ($IPD_{risk}$ = 0.2) and after lockdown ($IPD_{risk}$ = 0.4) reproduced the reported reduction in the annual IPD incidence while maintaining a stable prevalence of pneumococcal carriage during lockdown (**Table 1**, see scenario S3). By itself however, this scenario did not allow to observe an increase in antibiotic resistance.

When we combined reduced antibiotic prescribing and a reduced risk of developing an IPD with community azithromycin use in a proportion of COVID-19 infected individuals, which remains in the body for an additional 15.5 days after the last dose, in a single scenario, this scenario satisfied the observed trends in AMR (**Table 1**, see scenario S19). A similar outcome was observed in scenario S29 when adding a longer carriage duration of antibiotic resistant strains on top of this, however, in the absence of community azithromycin use in COVID-19 infected (**Table 1**, see scenario S1, S5, S3, S19, and S20) trends of increasing antibiotic resistance cannot be reproduced. Therefore, our best model scenario for describing the observed trends combined: (1) a reduction in the overall community antibiotic consumption; (2) the assumption that lockdown effectively reduced SARS-CoV-2 transmission including transmission of other respiratory viruses, but not pneumococcal carriage transmission, indirectly reducing the risk of developing an IPD; (3) either identical or longer carriage durations of antibiotic-resistant strains compared to antibiotic-sensitive strains, and (4) the community azithromycin use in a proportion of COVID-19 infected individuals.

## Effect of age

Next, we used the pandemic scenario S19 that best explains the reported trends to test the model using different parameter combinations to mimic different subpopulations (children and the elderly) considering that SARS-CoV-2 infection risk, pneumococcal disease risk, disease severity, bacterial carriage prevalence, and antibiotic prescribing are all highly heterogeneous across age groups. Using scenario S19, we initialized the model with lower and higher baseline carriage prevalence (10%, 20%, and 30%) (**Cohen et al., 2023**; **Rose et al., 2021**; **Rybak et al., 2022**; **Tinggaard et al., 2023**; **Wang et al., 2017**), we varied durations of pneumococcal carriage (20, 30, and 45 days), pneumococcal invasion rate, and considered reductions of antibiotic consumption at various levels (–13%, –18%, and

**Table 1.** Five mechanisms implemented in 31 pandemic scenarios proposed to explain the reported trends of IPD incidence, antibiotic resistance, and pneumococcal carriage in *S. pneumoniae*.

Scenarios explore all possible combinations of mechanisms proposed to test hypotheses that can explain the reported trends of annual invasive pneumococcal disease incidence (annual no. of cases per 100,000 inhabitants), antibiotic resistance (% of annual antibiotic-resistant IPD cases among total IPD cases), and % change in the pneumococcal carriage prevalence at the end of the first 60-day lockdown compared the prevalence before the lockdown. Model simulations were initiated assuming the initial 20% antibiotic resistance. Two pre-pandemic scenarios assume no SARS-CoV-2 circulation in the population and allow for the same 30-day carriage duration of both antibiotic-sensitive and -resistant strains ($d_S = d_R$) or a longer, 40-day carriage duration of -resistant strains ($d_R > d_S$). When implemented, these five mechanisms assume 18% reduction in community antibiotic prescribing, a reduced risk of developing an IPD during (0.2) and after the first lockdown (0.4), a 25% reduction in transmission of pneumococcal carriage during the first lockdown, a 10% of azithromycin use among COVID-19 infected individuals, and a longer 40-day carriage duration of -resistant strains. For a full list of parameters see *Appendix 2—table 2*. Reported trends in European countries showed a decrease in annual IPD incidence by 44.3% on average, an increase in antibiotic resistance, and generally stable asymptomatic pneumococcal carriage in healthy individuals during the first lockdown period. Only scenarios S19 and S29 fulfill all three reported trends during the COVID-19 pandemic in 2020 simultaneously while accounting for the reported reduction in community antibiotic prescribing ($d_S$ = carriage duration of antibiotic-sensitive pneumococcal strains; $d_R$ = carriage duration of antibiotic-resistant pneumococcal strains; PENI = penicillin; ERY = erythromycin).

| Scenarios | Mechanisms | | | | | IPD inc. | AR (%) | Sp. (%) |
|---|---|---|---|---|---|---|---|---|
| | 1 | 2 | 3 | 4 | 5 | | | |
| *Pre-pandemic 1: ($d_S = d_R$)* | | | | | | 10.8 | 20.0 | NA |
| *Pre-pandemic 2: ($d_R > d_S$)* | | | | | x | 11.3 | 20.0 | NA |
| *Pandemic: S1* | x | | | | | 10.9 | 19.2 | +1.3 |
| S2 | | x | | | | 8.9 | 20.1 | −36.1 |
| S3 | | | x | | | 5.9 | 20.0 | 0 |
| S4 | | | | x | | 9.9 | 23.7 | −9.1 |
| S5 | | | | | x | 11.3 | 20.0 | 0 |
| S6 | x | x | | | | 9.1 | 19.4 | −35.2 |
| S7 | x | | x | | | 6.0 | 19.4 | +1.3 |
| S8 | x | | | x | | 10.1 | 22.9 | −8.0 |
| S9 | x | | | | x | 11.5 | 19.3 | +1.3 |
| S10 | | x | x | | | 5.2 | 20.0 | −36.1 |
| S11 | | x | | x | | 8.9 | 20.1 | −36.1 |
| S12 | | x | | | x | 9.4 | 20.9 | −34.3 |
| S13 | | | x | x | | 5.6 | 22.5 | −9.1 |
| S14 | | | x | | x | 6.2 | 20.0 | 0 |
| S15 | | | | x | x | 10.4 | 23.4 | −9.1 |
| S16 | x | x | x | | | 5.3 | 19.6 | −35.2 |
| S17 | x | x | | x | | 8.3 | 22.4 | −41.3 |
| S18 | x | x | | | x | 9.6 | 20.3 | −33.5 |
| S19 | x | | x | x | | 5.7 | 22.0 | −8.0 |
| S20 | x | | x | | x | 6.3 | 19.5 | +1.3 |
| S21 | x | | | x | x | 10.6 | 22.7 | −7.9 |
| S22 | | x | x | x | | 5.0 | 22.0 | −42.1 |
| S23 | | x | x | | x | 5.5 | 20.6 | −34.3 |

*Table 1 continued on next page*

*Table 1 continued*

| | (1) | (2) | (3) | (4) | (5) | IPD inc. | AR (%) | Sp. (%) |
|---|---|---|---|---|---|---|---|---|
| *S24* | | x | | x | x | 8.7 | 23.9 | −40.2 |
| *S25* | | | x | x | x | 5.9 | 22.3 | −9.1 |
| *S26* | x | x | x | x | | 5.0 | 21.6 | −41.3 |
| *S27* | x | x | x | | x | 5.6 | 20.1 | −33.5 |
| *S28* | x | x | | x | x | 8.8 | 23.2 | −39.4 |
| *S29* | x | | x | x | x | 5.9 | 21.8 | −7.9 |
| *S30* | | x | x | x | x | 5.2 | 22.5 | −40.2 |
| *S31* | x | x | x | x | x | 5.3 | 22.1 | −39.4 |

| REPORTED TRENDS: | IPD inc. | AR (%) | Sp. (%) |
|---|---|---|---|
| Pre-pandemic (FR, 2019) | 10.5 [10.3–10.7] | 26.2 (PENI) and 20.9 (ERY) | NA |
| Pandemic (FR, 2020) | 5.8 [5.7–5.9] | 35.5 (PENI) and 23.0 (ERY) | Stable |
| *Pandemic (EU/EEA, 2020) General trends* | *Decrease by 44.3% on avg.* | *Majority of EU countries report an increase* | *Generally stable* |

Mechanisms: (1) Reduced community antibiotic prescribing; (2) Lockdown effect on reducing transmission of S. pneumoniae; (3) Reduced risk of developing an IPD; (4) Community azithromycin use in COVID-19 infected individuals; (5) Longer carriage duration of antibiotic-resistant pneumococcal strains.

−39%) consistent with the French data along with a range of community azithromycin use in COVID-19 infected (0–20%). For a full list of parameters see *Appendix 2—table 2*. Simulations showed that annual IPD incidence decreased between 43% and 51% compared to the pre-pandemic (baseline) scenario for children, the elderly, and the general population (*Figure 3*, grey bars). Although the overall antibiotic prescribing in the community was reduced (between 13% and 39%), antibiotic resistance is expected to increase (from 20.1% up to 23.6% in the elderly and from 32.8% up to 36.0% in children) compared to the pre-pandemic period in all age groups and in all scenarios where azithromycin was used in COVID-19 infected individuals (*Figure 3*, red bars). Daily prevalence of total pneumococcal carriage remained relatively stable, exhibiting higher levels of decrease with increased azithromycin use, while the prevalence of antibiotic-resistant pneumococcal carriage is expected to increase since clearance of antibiotic-susceptible strains due to azithromycin use shifts the competitive balance in favor of the existing resistant strains (*Figure 3*, third panel).

General trends produced in model simulations using scenario S19 remained unchanged across different age groups. The extent of the impact depended on the combined magnitude of a decrease in the general antibiotic use in the community and a degree of azithromycin use in COVID-19 infected individuals belonging to a particular age group or a subpopulation. In the elderly (≥65 years-old) and the general population, antibiotic resistance is expected to increase due to azithromycin use in COVID-19 infected. Black arrows indicate model outcomes that approximate the reported trends in antibiotic resistance in France for different age groups including general population (*Figure 3*). Only in instances when there was no azithromycin use in COVID-19 infected individuals, we observed a decrease in antibiotic resistance relative to the pre-pandemic period (e.g. children <5 years-old). When combining the largest decrease in overall antibiotic use with no or minimal azithromycin use in COVID-19 infected individuals, we expect to see the largest decrease or no change in antibiotic resistance relative to the pre-pandemic period.

## Effect of SARS-CoV-2 basic reproduction number ($R_0$) and within-host pathogen interactions on AMR

Considering that model simulations reproduced an absolute increase in antibiotic resistance comparable to that of 2% reported for macrolides in France but did not reproduce the reported larger increase in penicillin resistance, which was more than a 9% rise (35.5% relative increase) in France, we

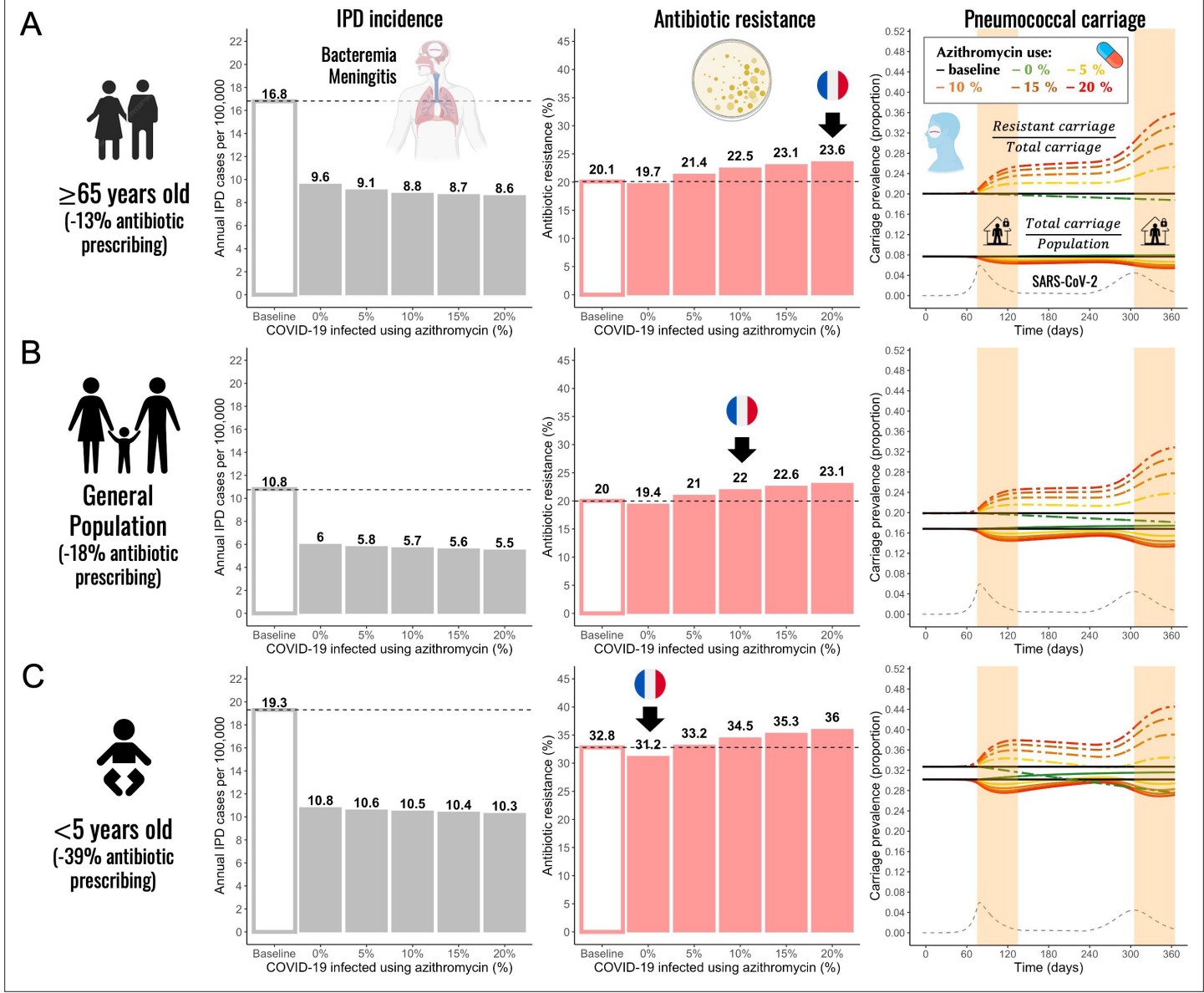

**Figure 3.** Annual incidence of invasive pneumococcal disease (IPD), antibiotic resistance (AR%), and pneumococcal carriage prevalence for three different subpopulations. (**A**) The elderly (≥65 years-old) (**B**) general population (all ages), and (**C**) children (<5 years-old). Using pandemic scenario S19, which includes a combination of three different mechanisms: reduced community antibiotic prescribing, a reduced risk of developing an IPD, and community azithromycin use in COVID-19 infected individuals, we ran model simulations for three different subpopulations. For a full list of parameter values see *Appendix 2—table 2*. Annual IPD incidence (grey bars) decreased between 43% and 51% relative to the pre-pandemic (baseline) period with magnitude of a decrease depending on an age group and the level of azithromycin use in COVID-19 infected individuals. Antibiotic resistance (red bars) increased compared to the pre-pandemic (baseline) period in all age groups whenever azithromycin was used in COVID-19 infected. Black arrows indicate model outcomes that approximate the reported trends in antibiotic resistance in France for different age groups. Daily prevalence of total pneumococcal carriage remained relatively stable (solid-colored lines), exhibiting higher levels of decrease with increased azithromycin use. The prevalence of antibiotic-resistant pneumococcal carriage increased (dashed colored lines) over time in relation to SARS-CoV-2 outbreak (black dashed line) and higher azithromycin use. Highlighted time intervals (days 75–135 and 305–365) represent two lockdown periods.

explored additional factors that may have amplified this increase. Using model scenario S19, we show that an association between higher values of SARS-CoV-2 $R_0$ and a greater percentage of COVID-19 infected individuals taking azithromycin leads to increased cumulative incidence of antibiotic-resistant IPDs and elevated antibiotic resistance (*Figure 4A*). For example, if pre-lockdown $R_0$ of SARS-CoV-2 was 3.8 instead of 3.2, model simulations predict an increase of 3.5% (23.5%) in antibiotic resistance

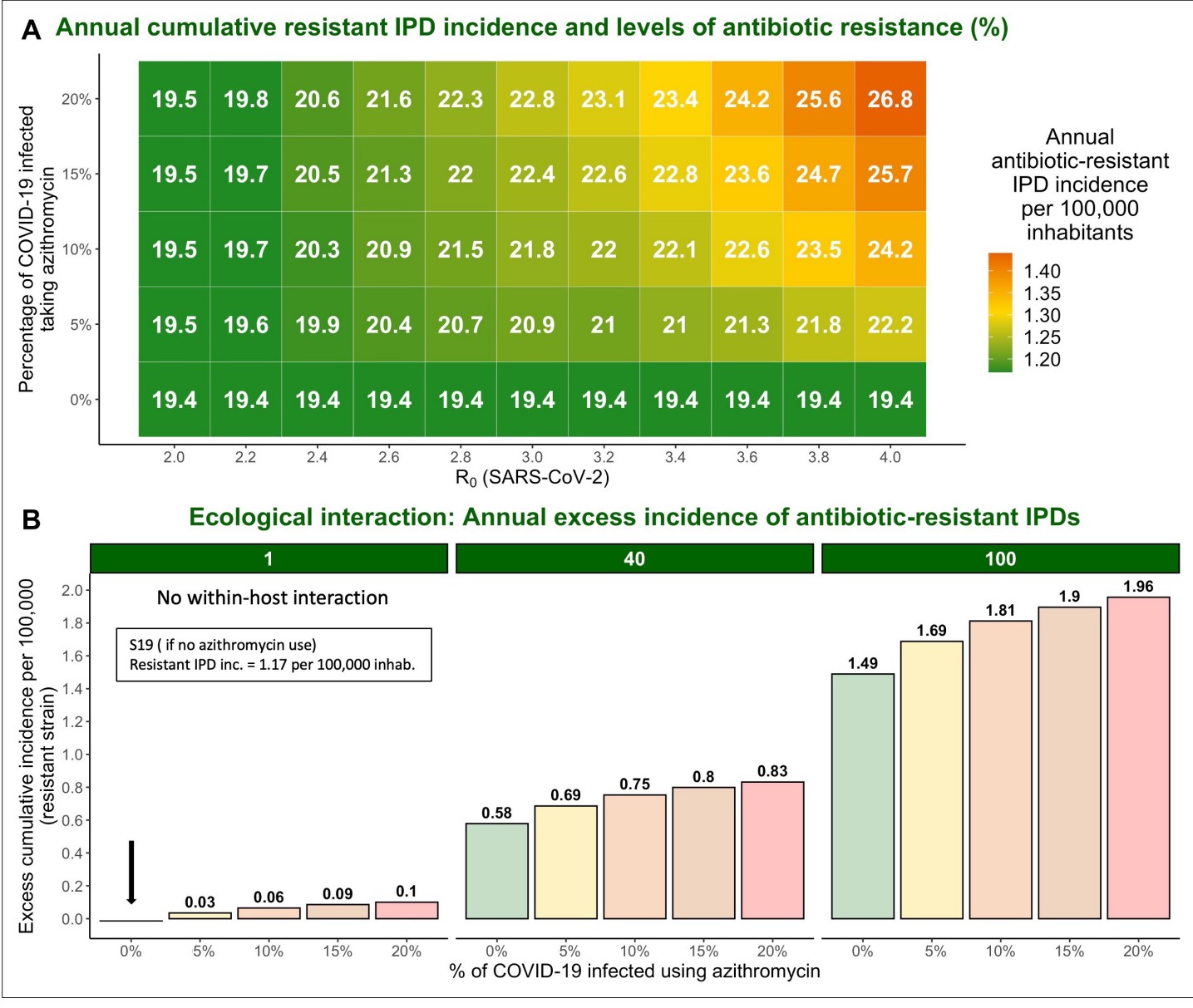

**Figure 4.** The impact of varying SARS-CoV-2 $R_0$ and percentage of COVID-19 infected individuals taking azithromycin in scenario S19 on antibiotic resistance (%) and the annual incidence of antibiotic-resistant invasive pneumococcal disease (IPD). Hypothetical within-host interactions contribute to an excess incidence of antibiotic-resistant IPDs. (**A**) Cumulative incidence of antibiotic-resistant IPDs and antibiotic resistance increase with greater values of SARS-CoV-2 $R_0$ and higher percentage of the COVID-19 infected individuals taking azithromycin. The reproduction number for SARS-CoV-2 ($R_0$) in the community corresponds to the most common estimates of $R_0$ in France and other European countries ranging from $R_0$=2–4 (**Allieta et al., 2022**; **D'Arienzo and Coniglio, 2020**; **Di Domenico et al., 2020**; **Flaxman et al., 2020**; **Liu et al., 2020**; **Roux et al., 2020**; **Salje et al., 2020**).
(**B**) Annual excess in cumulative antibiotic-resistant IPD incidence in scenario S19 due to synergistic within-host ecological interactions compared to the same scenario with no within-host interactions and no azithromycin use (1.17 resistant IPD cases/100,000 inhabitants). A rate of disease progression increased by a factor $\psi_c = 1$ (no within-host interaction) and $\psi_c = 40$ in scenario S19 applied to the general population assuming azithromycin use in 10% of the infected individuals resulted in approximately 0.06 and 0.75 additional cases of antibiotic-resistant disease per 100,000 inhabitants over the course of 1 year, respectively, compared to the scenario S19 assuming no within-host interaction and no azithromycin use (indicated by the black arrow). For more details, see **Appendix 2—figure 1**.

from the pre-pandemic levels instead of 2%. As the $R_0$ value increases, the impact of azithromycin use becomes more pronounced.

Assuming within-host interactions where SARS-CoV-2 infection favors progression from pneumococcal colonization to disease ($\psi_c > 1$), we found that surges in COVID-19 cases accompanied by increasing levels of azithromycin use lead to excess number of cases caused by antibiotic-resistant

strains. Indeed, a rate of disease progression increased by a factor $\psi_c = 40$ in scenario S19 with 10% of infected using azithromycin applied to the general population results in approximately 0.75 additional cases of antibiotic-resistant disease per 100,000 inhabitants over the course of 1 year compared to 0.06 additional cases if there are no within-host interactions (*Figure 4B*). This represents 5% rise in resistance from the pre-pandemic levels (25% relative increase).

## Discussion

We propose a novel co-circulation model describing the spread of SARS-CoV-2 and antibiotic-resistant bacteria in a community setting to show how human behavioral responses to the COVID-19 pandemic can differentially impact antibiotic resistance. Our model simulations assessed different hypotheses proposed to explain the observed trends of antibiotic resistance, IPD incidence, and pneumococcal carriage. We identified the most plausible mechanisms underlying the observed patterns of resistance and disease incidence, showing how lockdowns indirectly substantially reduce the incidence of IPD, while surges in COVID-19 cases accompanied by antibiotic prescribing in COVID-19 infected individuals increase antibiotic resistance.

Many studies have reported trends on the incidence of community-acquired bacterial infections since the onset of the pandemic (*Brueggemann et al., 2021*; *Shaw et al., 2023*). There was a significant reduction in the risk of invasive disease caused by *S. pneumoniae* (risk ratio 0·47; 95% CI 0·40–0·55; *Shaw et al., 2023*). Initially, this observation seemed to support the hypothesis that NPIs implemented to control SARS-CoV-2 transmission may have simultaneously reduced the incidence of bacterial infections by preventing bacterial transmission and acquisition (*Brueggemann et al., 2021*; *Kadambari et al., 2022*). Indeed, the scenario of lockdown impact on pneumococcal transmission reproduced such trends. However, incorporating a mechanism of reduced risk for developing an IPD due to the absence of circulation of common respiratory viruses led to similar estimates of the relative reduction in IPD incidence as reported in the EU/EEA for 2020 (*Brueggemann et al., 2021*; *European Centre for Disease Prevention and Control and World Health Organization, 2022*). This finding, coupled with the outcome of other studies that found a generally stable pneumococcal carriage prevalence in healthy individuals, both children and adults, during COVID-19 containment measures (*Nation et al., 2023*; *Petrović et al., 2022*; *Rybak et al., 2022*; *Willen et al., 2021*; *Wyllie et al., 2023*), supports the alternative hypothesis. This explanation accounts for the decreased incidence of IPD, rather than attributing it to reduced pneumococcal transmission, which resulted in a significant reduction in carriage according to the simulations (*Smith and Opatowski, 2021*). Furthermore, a study in Vietnam found that reductions in IPD associated with NPIs may be due to reductions in overall pneumococcal carriage density rather than carriage prevalence, driven by reductions in capsular pneumococcal carriage density frequently implicated in IPD (*Nation et al., 2023*). Considering that common respiratory viruses such as influenza increase pneumococcal carriage density, which contributes to transmission and disease, this hypothesis seems plausible (*Alpkvist et al., 2015*; *Diavatopoulos et al., 2010*; *McCullers et al., 2010*; *Short et al., 2012*; *Wolter et al., 2014*).

Globally, community antibiotic consumption dropped during the first year of the COVID-19 pandemic compared to the pre-pandemic period. Decreasing temporal trends were observed in England (*Hussain et al., 2021*), Canada (*Mamun et al., 2021*), the United States (*Buehrle et al., 2021*), China (*Zhang et al., 2021*), South Korea (*Ryu et al., 2021*), New Zealand (*Duffy et al., 2021*), and across European countries (*Högberg et al., 2021*). In France in particular, the number of antibiotic prescriptions decreased by 18.2% in the general population; however, this reduction ranged from 13% to 39% for the oldest and youngest age groups, respectively (*Bara et al., 2022*). These trends in antibiotic prescribing may largely be explained by reduced incidence of seasonal respiratory tract infections and reduced primary care consultations (*Andrews et al., 2022*; *Homeniuk and Collins, 2021*). On the other hand, the advent of telemedicine, pandemic-induced patient stress, and increased antibiotic demand may have partly offset prescription reductions due to decreased consultations and healthcare-seeking behavior (*Hsu, 2020*; *Read et al., 2023*). In a global analysis of antimicrobial sales, Khouja et al. found that antibiotic consumption initially increased by approximately 7% in March 2020, prior to subsequent declines through to August 2020 (*Khouja et al., 2022*). While overall antibiotic prescribing may have decreased, prescription of specific antibiotics has increased, particularly those associated with COVID-19 patient management. Across continents, a rise of 10% in

monthly COVID-19 cases exhibited a correlative trend with elevated macrolide sales of 0.8%, 1.3%, and 1.5% in Europe, North America, and Africa, respectively (*Nandi et al., 2023*).

Community consumption of azithromycin, a macrolide, increased during the first year of the pandemic in multiple countries with significant variation across geographic locations and with greatest prescribing among older patients (*Bara et al., 2022*; *Bednarčuk et al., 2023*; *Bogdanić et al., 2022*; *Crisafulli et al., 2021*; *Parveen et al., 2020*; *Weill et al., 2021*). In an outpatient setting in southern Italy between February 2020 and January 2021, azithromycin represented 42.1% of all drug prescriptions to individuals diagnosed with COVID-19, while all other antibiotics combined represented just 20.9% (*Crisafulli et al., 2021*). A study in northwest London across two epidemic waves between January and August 2020 found that, among COVID-19 patients prescribed an antibiotic by a general practitioner during the study period, 31.5% received their prescription within 14 days of a positive SARS-CoV-2 test (*Zhu et al., 2021*). Two large USA-based studies have also described early pandemic antibiotic prescribing among COVID-19 patients. From April 2020 to April 2021, approximately 30% of outpatient COVID-19–related visits among Medicare beneficiaries (≥65 years-old) have resulted in a filled antibiotic prescription, 50.7% of which were for azithromycin (*Tsay et al., 2022*). For 0-to-5 year-olds and 45-to-64 year-olds, 4% and 16% of outpatient COVID-19–related visits have resulted in a filled antibiotic prescription, respectively (*Wittman et al., 2023*). In the Alsace region in France, there was a clear peak azithromycin prescribing during the first wave of the COVID-19 (*Danion et al., 2023*). During the first lockdown in France, community azithromycin consumption increased by 25.9%, with the increase varying from 13.4% to 47.3% depending on the week (*Weill et al., 2021*), while the overall number of azithromycin prescriptions across France in 2020 increased by 10.1% relative to 2019 (*Bara et al., 2022*). Azithromycin treatment usually lasts 3–5 days depending on the disease, but the drug stays in the system for about 15.5 days after the last dose due to the long half-life of more than 60 hr (*Foulds et al., 1990*; *Girard et al., 2005*). On the other hand, penicillin has an elimination half-life of approximately 1.4 hr and leaves the body in 7.7 hr after the last dose. This suggests that if azithromycin consumption increased during the first year of the pandemic, antibiotic exposure time also increased as a result, although the overall number of antibiotic prescriptions decreased. Moreover, the use of azithromycin has been associated with selection of both macrolide and non-macrolide resistance (*Doan et al., 2020*). In a study investigating the direct effect of antibiotic exposure on resistance in the oral streptococcal flora of healthy volunteers, use of azithromycin (500 mg once daily for 3 days) significantly increased the proportion of macrolide-resistant streptococci in healthy individuals (*Malhotra-Kumar et al., 2007*). Resistance peaked at day four in the azithromycin group and this increase remained significantly higher in the azithromycin group than in the placebo group until day 180 (*Malhotra-Kumar et al., 2007*). A clinical trial of mass azithromycin distributions for treating trachoma in Ethiopia resulted in an increase in resistant *S. pneumoniae* isolates among children under the age of 10 (*Keenan et al., 2018*; *Keenan et al., 2015*).

Our model simulations show that antibiotic resistance increases with surges in SARS-CoV-2 infections when there is a corresponding increase in azithromycin use, but that lockdowns can moderate this increasing trend by effectively limiting transmission of SARS-CoV-2 (*Salje et al., 2020*). Conversely, surges in azithromycin prescribing during SARS-CoV-2 outbreaks in the absence of effective measures to prevent transmission, as reported in certain regions and pandemic periods, may cause substantial increases in antibiotic resistance. Our model successfully captured the main trends of antibiotic resistance and IPD incidence observed in Europe in 2020 for *S. pneumoniae*. However, not all European countries reported an increase in antibiotic resistance. This inter-country heterogeneity may not be due only to heterogeneity of antibiotic use as shown in our model but may be attributed to other pandemic factors not directly implemented or assumed in the model scenario, such as different adherence to COVID-19 control measures across countries and different age groups, including impacts on disease surveillance and data reporting during the pandemic. Real-life scenarios are significantly more complicated and involve multiple alterations of many pandemic factors at different points in time and heterogeneity across populations (e.g. antibiotic prescribing increases in some demographic groups and decreases in others, multiple lockdowns, curfews, or telework).

In our model simulations, we used SARS-CoV-2 parameter value $R_0=3.2$ (*Liu et al., 2020*) in the absence of population immunity, best reflecting epidemiological dynamics from early in the pandemic. The most common estimates of SARS-CoV-2 $R_0$ in France and other European countries ranged from $R_0=2–4$ (*Flaxman et al., 2020*; *Liu et al., 2020*). Modeling results suggest that higher SARS-CoV-2

$R_0$ estimates combined with higher proportion of COVID-19 infected individuals using azithromycin exacerbated impacts of COVID-19 on antibiotic resistance (*Figure 4A*). However, the overall impacts of COVID-19 on AMR are difficult to predict, likely vary over the short, medium, and long term, and depend on the organism, setting, and subpopulation considered.

SARS-CoV-2 bacterial coinfection has been reported relatively rarely over the course of the pandemic, suggesting that most COVID-19 patients probably do not require antibiotic therapy (*Garcia-Vidal et al., 2021*; *Karami et al., 2021*; *Langford et al., 2020*), although extensive prophylactic antibiotic use may have limited observed co-infection incidence. The inflammatory immune response resulting from COVID-19 likely predisposes patients to subsequent progression to an IBD to some extent (*Sender et al., 2021*), but antibiotic use may also favor progression to IBD for patients colonized with drug-resistant strains (*Baggs et al., 2018*). We do not explicitly model the dynamics of interaction since strong evidence for such interactions remains limited (*Wong et al., 2023*). The results presented in *Figure 4B* suggest that such within-host interactions could have important consequences for the resistant IPD incidence during COVID-19 waves, especially in the elderly and high-risk groups. The model's structure allows for easy integration of mechanistic interactions as more information becomes available on this phenomenon.

Our study focused on the general community, but COVID-19 distinctly influenced AMR in hospitals and long-term care facilities. Extensive antibiotic use in COVID-19 patients and disruptions to antibiotic stewardship programs may have increased antibiotic-resistant carriage in these settings. A meta-analysis conducted on studies published until June 2020 found that 68–81% of hospitalized COVID-19 patients and 74–94% in intensive care received antibiotics (*Monnet and Harbarth, 2020*). The disorganization in hospitals during the COVID-19 pandemic might have reduced antibiotic resistance surveillance, allowing resistant organisms to spread. However, the early implementation of antibiotic stewardship programs in March 2020, patient isolation, and widespread use of personal protective equipment (PPE) have mitigated this increase to some degree (*Henig et al., 2021*; *Monnet and Harbarth, 2020*; *Seaton et al., 2020*; *Van Laethem et al., 2021*). Models analyzing these impacts in hospitals contribute to understanding COVID-19's specific role in the antibiotic resistance burden in different settings (*Smith et al., 2023*).

A limitation of our model is the lack of age structure and contact patterns between age groups, as SARS-CoV-2 infection risk, pneumococcal disease risk, disease severity, bacterial carriage prevalence and antibiotic prescribing are all highly heterogeneous across age groups. While this choice was made to keep the model as simple as possible, we tested the model using different parameter combinations to mimic different subpopulations (children and ≥65 years-old). This included varying durations of pneumococcal carriage, initializing the model with lower and higher baseline carriage prevalence, considering reductions of general antibiotic consumption at various levels, and varying a percentage of COVID-19 infected individuals using azithromycin. Simulations of the different age groups individually interestingly reproduced realistic trends by age.

In conclusion, we introduce the first epidemiological model outlining the impact of the COVID-19 pandemic on the dynamics of AMR in the community. Our work demonstrates the utility of mathematical modeling approach in unraveling the complex effects of the COVID-19 pandemic responses AMR dynamics. While our model was structured and parameterized based upon *S. pneumoniae*, its adaptability allows for application to various bacteria and epidemiological scenarios in the community (e.g. impacts of SARS-CoV-2-bacteria interactions in the context of seasonal outbreaks of endemic pathogens). Future research would benefit from fitting the model to real-world data for different bacterial species to enhance our understanding of AMR trends.

## Methods

### *Streptococcus pneumoniae* surveillance data

Antibiotic resistance trends reported in 2019 and 2020, provided by EARS-Net (European Antimicrobial Resistance Surveillance Network) were acquired from a joint 2022 report on AMR during 2020 by WHO and ECDC (*European Centre for Disease Prevention and Control and World Health Organization, 2022*). The annual incidence of *S. pneumoniae* invasive isolates for 2019 and 2020 was measured as the number of invasive isolates from blood or cerebrospinal fluid. The proportion of resistant isolates represents the proportion of isolates with phenotypic resistance to penicillin and

macrolides using standardized bacterial culture methods and EUCAST breakpoints. Out of 28 European countries that reported antibiotic resistance data for *S. pneumoniae*, 24 countries had enough samples to establish 2019–2020 resistance trends for penicillin and macrolides. The resistance data for France, which were subsequently analyzed, were provided by the CNRP (The French National Reference Center for Pneumococci).

## Model structure

We developed a pathogen co-circulation model (*Appendix 2—figure 2*) written using systems of ordinary differential equations (ODEs) (Appendix 2-Equations; code available on GitHub, copy archived at *Kovacevic, 2024*). The model simultaneously describes potential infection with SARS-CoV-2 and colonization with antibiotic-sensitive and/or -resistant strains of *S. pneumoniae* in a well-mixed community population. SARS-CoV-2 infection is modeled by a Susceptible-Exposed-Infectious-Recovered (SEIR) process where individuals become exposed to SARS-CoV-2 at rate $\beta_C$ upon contact with other infected individuals. Infection begins with a non-infectious exposed period lasting $1/\varepsilon$ days and is followed by an infectious period lasting $1/\gamma^C$ days, eventually leading to recovery and immunization against future re-infection. Waning immunity and competitive multi-strain SARS-CoV-2 dynamics are not considered.

Individuals in S, E, I, and R compartments can be uncolonized with *S. pneumoniae* (U), colonized with either a drug-sensitive ($C^S$) or a drug-resistant strain ($C^R$), or co-colonized with two strains ($C^{SS}$, $C^{RR}$, $C^{SR}$). Colonization with each respective strain is acquired at rates $\beta_S$ and $\beta_S f$ upon contact with other colonized individuals (*Appendix 2—table 2*). We assume a metabolic cost of resistance, whereby the drug-resistant strain has a reduced intrinsic transmission rate relative to the drug-sensitive strain due to reduced fitness, *f*. Bacterial carriage is cleared naturally after an average duration of $\frac{1}{\gamma^S} = \frac{1}{\gamma^R} = \frac{1}{\gamma^{SR}} = \frac{1}{\gamma^{SS}} = \frac{1}{\gamma^{RR}}$ days, which we assume to be the same for all types of carriers in our baseline scenario (in the scenarios assuming longer carriage duration of antibiotic-resistant strains, $\frac{1}{\gamma^S} = \frac{1}{\gamma^{SS}}$ and $\frac{1}{\gamma^R} = \frac{1}{\gamma^{SR}} = \frac{1}{\gamma^{RR}}$). We further assume that some share of the population is exposed to antibiotics at any given time, independent of bacterial carriage, with individuals initiating antibiotic therapy at rate $\tau$, which lasts for an average duration of $\frac{1}{r}$ days. Another model assumption is that a proportion $p_{az}$ of those infected with COVID-19 in the community (between 0% and 20% of individuals in an I compartment) receive azithromycin prescription from general practitioner reflecting azithromycin prescriptions in the early pandemic, while the rest of the infectious individuals $(1 - p_{az})$ are exposed to the baseline antibiotic therapy. We assume baseline treatment duration of 7 days, on average, regardless of the antibiotic prescribed and without any assumed persistence of the antibiotic in the system after the last dose $\frac{1}{r}$. In case of antibiotic treatment with azithromycin for COVID-19 infected individuals we assume the treatment lasts three days with antibiotics remaining in the system for additional 15.5 days after the last dose for a total of 18.5 days of antibiotic exposure where COVID-19 recovered individuals $R_{az}$ treated with azithromycin retain azithromycin in their system for an additional 11.5 days $\frac{1}{r_{az}}$ after COVID-19 recovery. Individuals treated with antibiotics are unable to acquire the sensitive strain. Antibiotics are assumed to clear colonization with sensitive strains at a rate $\omega$ while having no direct impact on colonization with resistant strains. This bacterial colonization process results in antibiotic selection for resistance via competition for limited hosts, facilitates epidemiological coexistence between strains and is adapted from previous models of *S. pneumoniae* (*Colijn et al., 2010*; *Lipsitch et al., 2009*; *Mulberry et al., 2020*). For a full list of parameter values see *Appendix 2—table 2*.

## Simulation in an early COVID-19 pandemic context

ODEs were integrated numerically using the R package deSolve to simulate and quantify epidemiological dynamics (*Soetaert et al., 2010*). First, bacterial dynamics were simulated until endemic equilibrium was achieved, under the assumption that *S. pneumoniae* was at endemic equilibrium upon the emergence of COVID-19. Second, using equilibrium states as initial conditions and re-initializing simulation time to *t*=0, a single SARS-CoV-2 infected individual was introduced into the population and ODEs were again integrated numerically to *t*=365 days. Parameter values used for simulation were taken from prior studies prioritizing French data and are provided in *Appendix 2—table 2*.

These simulations were conducted in the context of an 'early pandemic scenario' coinciding with the implementation of population-wide NPIs to slow SARS-CoV-2 transmission. This was conceived as the implementation of two 60 day lockdown periods starting on day 75 and on day 305 in response to the simulated surge in COVID-19 cases. Lockdowns were assumed to have three major potential

impacts on population behavior and, in turn, the transmission dynamics of SARS-CoV-2 and *S. pneumoniae*. These impacts were incorporated into simulations by modifying epidemiological parameters in the model coincident with lockdowns. Three such modifications were considered and switched on and off, considering all possible combinations. First, lockdown led to reduced SARS-CoV-2 transmissibility by a factor $\theta_c$. Second, lockdown led to a population-wide change in antibiotic initiation rate by a factor $a$ (representing modified healthcare-seeking behavior leading to a reduction in the number of antibiotic prescriptions). Finally, lockdowns changed the pneumococcal disease risk by a factor $IPD_{risk}$ (representing a reduced risk of developing an IPD due to the absence of other respiratory viruses).

### Effect of SARS-CoV-2 basic reproduction number ($R_0$) on AMR

Impacts of SARS-CoV-2 on antibiotic-resistant IPD incidence may also depend on the characteristics of locally circulating SARS-CoV-2 $R_0$. To account for potential impacts of SARS-CoV-2 transmissibility and azithromycin use in the community, in simulations we varied (i) values of $R_0$ (basic reproduction number) for SARS-CoV-2 in France ($2 \leq R_0 \leq 4$) and (ii) the proportion of the COVID-19 infected individuals using azithromycin at simulation outset (from 0% to 20%).

### Effect of within-host interactions on AMR

SARS-CoV-2 infection may impact progression from bacterial colonization to IBD at the within-host level. To incorporate this mechanism in our model, we included a within-host interaction term in scenario S19: the ecological interaction term $(\psi_c)$ increases the rate of progression to invasive disease among colonized individuals who are also infected with SARS-CoV-2. The equations for calculating daily IPD incidence assuming within-host interactions due to SARS-CoV-2 co-infection with accompanying details can be found in Appendix 2.

## Additional information

### Competing interests

Robert Cohen: received consulting fees from Pfizer, Sanofi, MSD, and GSK, including travel grants from Pfizer and payments from Symposium Pfizer, MSD, GSK, and Sanofi, and participated in advisory/data safety monitoring board at Pfizer, Sanofi, MSD, and GSK. Corinne Levy: received travel grants from Pfizer and payments from Symposium Pfizer and MSD. Lulla Opatowski: received a research grant from Pfizer and Sanofi Pasteur on unrelated topics through her institution. The other authors declare that no competing interests exist.

### Funding

| Funder | Grant reference number | Author |
| --- | --- | --- |
| Agence Nationale de la Recherche | ANR-10-LABX-62 IBEID | Aleksandra Kovacevic Lulla Opatowski |
| Université Paris-Saclay | AAP Covid-19 2020 | Lulla Opatowski |
| Agence Nationale de la Recherche | SPHINX-17-CE36-0008-01 | Laura Temime |
| Fondation de France | MODCOV Grant 106059 | Lulla Opatowski Laura Temime |
| Institut National de la Santé et de la Recherche Médicale | | Lulla Opatowski |
| Institut Pasteur | | Lulla Opatowski |
| Conservatoire National des Arts et Métiers | | Laura Temime |
| Université de Versailles Saint-Quentin-en-Yvelines | | Lulla Opatowski |

| Funder | Grant reference number | Author |
|---|---|---|

The funders had no role in study design, data collection and interpretation, or the decision to submit the work for publication.

## Author contributions

Aleksandra Kovacevic, Conceptualization, Data curation, Formal analysis, Investigation, Visualization, Methodology, Writing – original draft, Writing – review and editing; David RM Smith, Conceptualization, Formal analysis, Investigation, Visualization, Methodology, Writing – original draft, Writing – review and editing; Eve Rahbé, Paul Henriot, Investigation, Writing – original draft, Writing – review and editing; Sophie Novelli, Investigation, Writing – original draft; Emmanuelle Varon, Robert Cohen, Corinne Levy, Writing – review and editing; Laura Temime, Conceptualization, Funding acquisition, Methodology, Writing – review and editing; Lulla Opatowski, Conceptualization, Supervision, Funding acquisition, Methodology, Writing – review and editing

## Author ORCIDs

Aleksandra Kovacevic (iD) https://orcid.org/0000-0001-9740-6207
Eve Rahbé (iD) http://orcid.org/0000-0002-3828-3910
Laura Temime (iD) http://orcid.org/0000-0002-8850-5403

## Decision letter and Author response
Decision letter https://doi.org/10.7554/eLife.85701.sa1
Author response https://doi.org/10.7554/eLife.85701.sa2

# Additional files

## Supplementary files
• MDAR checklist

## Data availability
Source Data 1 contains the numerical data used to generate Figure 1.

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

## Appendix 1

### Literature search

We searched PubMed on December 4, 2023, for published mathematical modeling studies describing coinfection with SARS-CoV-2 and antibiotic-resistant bacteria and including epidemiological transmission of both pathogens. We used the following search query:

("model*"[Title/Abstract] OR "models, theoretical"[MeSH Terms] OR "simulat*"[Title/Abstract]) AND

("mathematic*"[MeSH Terms] OR "compartmental"[Title/Abstract] OR "agent-based"[Title/Abstract] OR "individual-based"[Title/Abstract] OR "stochastic*"[Title/Abstract] OR "deterministic"[Title/Abstract] OR "computer-generated"[Title/Abstract] OR "computer simulation"[MeSH Terms] OR "computer"[Title/Abstract] OR "computational"[Title/Abstract] OR "mathematical*"[Title/Abstract] OR "theoretical"[Title/Abstract] OR "prediction"[Title/Abstract] OR "simulat*"[Title/Abstract])

AND

("outbreak*"[Title/Abstract] OR "dynamic*"[Title/Abstract] OR "disseminat*"[Title/Abstract] OR "propagat*"[Title/Abstract] OR "transmi*"[Title/Abstract] OR "cross infection/transmission"[MeSH Terms] OR "spread*"[Title/Abstract] OR "diffus*"[Title/Abstract] OR "circulati*"[Title/Abstract] OR "disease activity"[Title/Abstract] OR "epidemiology"[Title/Abstract] OR "co-infection"[Title/Abstract])

("COVID-19"[Title/Abstract] OR "SARS-CoV-2"[Title/Abstract])

("bacteria*"[Title/Abstract] OR "antibiotic resist*"[Title/Abstract] OR "antibiotic-resist*"[Title/Abstract] OR "antimicrobial resist*"[Title/Abstract] OR "antimicrobial-resist*"[Title/Abstract]),

which returned 153 papers. One published study developed a mathematical model including the transmission of both SARS-CoV-2 and antibiotic-resistant bacteria but is specific to the hospital setting (*Smith et al., 2023*). Two bacterial transmission models implicitly accounted for COVID-19 by simulating pandemic-associated social distancing interventions/lockdowns, and corresponding reductions in (i) close-proximity contacts, in the context of transmission of antibiotic-resistance genes among human microbiota (*Rebelo et al., 2021*), and (ii) sexual contacts, in the context of transmission of HIV, gonorrhea and chlamydia (*Jenness et al., 2021*). A third model of *Neisseria meningitidis* transmission also evaluated colonization and infection dynamics over the pandemic period (*Cascante-Vega et al., 2023*). However, none of these models included SARS-CoV-2 infection and transmission, and only the first considered antibiotic resistance. One further transmission modelling study included coinfection with SARS-CoV-2 and influenza-like illness but did not consider bacteria (*Bhowmick et al., 2023*). Our search also returned some statistical models used to analyze epidemiological trends in bacterial coinfection subsequent to SARS-CoV-2 infection (*Zhou et al., 2023*) and changes in the incidence of bacterial disease during the pandemic (*Shaw et al., 2023*), but these did not use transmission modelling approaches to understand or quantify the underlying epidemiological mechanisms driving these trends. Several other modelling studies have described within-host molecular dynamics or airborne fluid particle dynamics, but do not describe pathogen transmission at the host population level. No other relevant articles were identified.

## Appendix 2

## Model equations:

Compartment notation:

$$X_j^i$$

*X*=SARS-CoV-2 status (S=susceptible, E=exposed, I=infected, and *R*=recovered)

*i*=bacterial carriage (no superscript =no carriage, S=antibiotic-sensitive carriage, *R*=antibiotic-resistant carriage, SR =dual carriage, SS =dual antibiotic-sensitive carriage, RR =dual antibiotic-resistant carriage)

*j*=antibiotic exposure (no subscript =no antibiotic exposure, a=exposed to baseline antibiotic treatment, az =exposed to azithromycin treatment)

Total population size:

$$
\begin{aligned}
N = \quad & S + S^S + S^R + S^{SR} + S^{SS} + S^{RR} + S_a + S_a^S + S_a^R + S_a^{SR} + S_a^{SS} + S_a^{RR} + E + E^S + E^R \\
& + E^{SR} + E^{SS} + E^{RR} + E_a + E_a^S + E_a^R + E_a^{SR} + E_a^{SS} + E_a^{RR} + I + I^S + I^R + I^{SR} \\
& + I^{SS} + I^{RR} + I_a + I_a^S + I_a^R + I_a^{SR} + I_a^{SS} + I_a^{RR} + I_{az} + I_{az}^S + I_{az}^R + I_{az}^{SR} + I_{az}^{SS} \\
& + I_{az}^{RR} + R + R^S + R^R + R^{SR} + R^{SS} + R^{RR} + R_a + R_a^S + R_a^R + R_a^{SR} + R_a^{SS} + R_a^{RR} \\
& + I_a^{RR} + R_{az} + R_{az}^S + R_{az}^R + R_{az}^{SR} + R_{az}^{SS} + R_{az}^{RR}
\end{aligned}
$$

Forces of infection:

*S. pneumoniae*

$$
\begin{aligned}
\lambda_S = \quad & \beta_S \theta_\beta \Big( S^S + 2qp_s S^{SR} + 2q S^{SS} + S_a^S + 2qp_s S_a^{SR} + 2q S_a^{SS} + E^S + 2qp_s E^{SR} + 2q E^{SS} + E_a^S \\
& + 2qp_s E_a^{SR} + 2q E_a^{SS} + I^S + 2qp_s I^{SR} + 2q I^{SS} + I_a^S + 2qp_s I_a^{SR} + 2q I_a^{SS} + I_{az}^S \\
& + 2qp_s I_{az}^{SR} + 2q I_{az}^{SS} + R^S + 2qp_s R^{SR} + 2q R^{SS} + R_a^S + 2qp_s R_a^{SR} + 2q R_a^{SS} + R_{az}^S \\
& + 2qp_s R_{az}^{SR} + 2q R_{az}^{SS} \Big) / N
\end{aligned}
$$

$$
\begin{aligned}
\lambda_R = \quad & \beta_S f \theta_\beta \Big( S^R + 2q \left(1 - p_s\right) S^{SR} + 2q S^{RR} + S_a^R + 2q \left(1 - p_s\right) S_a^{SR} + 2q S_a^{RR} + E^R \\
& + 2q \left(1 - p_s\right) E^{SR} + 2q E^{RR} + E_a^R + 2q \left(1 - p_s\right) E_a^{SR} + 2q E_a^{RR} + I^R + 2q \left(1 - p_s\right) I^{SR} \\
& + 2q I^{RR} + I_a^R + 2q \left(1 - p_s\right) I_a^{SR} + 2q I_a^{RR} + I_{az}^R + 2q \left(1 - p_s\right) I_{az}^{SR} + 2q I_{az}^{RR} + R^R \\
& + 2q \left(1 - p_s\right) R^{SR} + 2q R^{RR} + R_a^R + 2q \left(1 - p_s\right) R_a^{SR} + 2q R_a^{RR} + R_{az}^R \\
& + 2q \left(1 - p_s\right) R_{az}^{SR} + 2q R_{az}^{RR} \Big) / N
\end{aligned}
$$

$$
\begin{aligned}
\lambda_{XS} = \quad & \beta_S \theta_\beta \Big( S^S + p_{single} \left(2qp_s S^{SR} + 2q S^{SS}\right) + S_a^S + p_{single} \left(2qp_s S_a^{SR} + 2q S_a^{SS}\right) + E^S \\
& + p_{single} \left(2qp_s E^{SR} + 2q E^{SS}\right) + E_a^S + p_{single} \left(2qp_s E_a^{SR} + 2q E_a^{SS}\right) + I^S \\
& + p_{single} \left(2qp_s I^{SR} + 2q I^{SS}\right) + I_a^S + p_{single} \left(2qp_s I_a^{SR} + 2q I_a^{SS}\right) + I_{az}^S \\
& + p_{single} \left(2qp_s I_{az}^{SR} + 2q I_{az}^{SS}\right) + R^S + p_{single} \left(2qp_s R^{SR} + 2q R^{SS}\right) + R_a^S \\
& + p_{single} \left(2qp_s R_a^{SR} + 2q R_a^{SS}\right) + R_{az}^S + p_{single} \left(2qp_s R_{az}^{SR} + 2q R_{az}^{SS}\right) \Big) / N
\end{aligned}
$$

$$
\begin{aligned}
\lambda_{XR} = \quad & \beta_S f \theta_\beta \Big( S^R + p_{single} \left( 2q\left(1-p_s\right) S^{SR} + 2q S^{RR} \right) + S_a^R \\
& + p_{single} \left( 2q\left(1-p_s\right) S_a^{SR} + 2q S_a^{RR} \right) + E^R + p_{single} \left( 2q\left(1-p_s\right) E^{SR} + 2q E^{RR} \right) \\
& + E_a^R + p_{single} \left( 2q\left(1-p_s\right) E_a^{SR} + 2q E_a^{RR} \right) + I^R \\
& + p_{single} \left( 2q\left(1-p_s\right) I^{SR} + 2q I^{RR} \right) + I_a^R + p_{single} \left( 2q\left(1-p_s\right) I_a^{SR} + 2q I_a^{RR} \right) \\
& + I_{az}^R + p_{single} \left( 2q\left(1-p_s\right) I_{az}^{SR} + 2q I_{az}^{RR} \right) + R^R \\
& + p_{single} \left( 2q\left(1-p_s\right) R^{SR} + 2q R^{RR} \right) + R_a^R + p_{single} \left( 2q\left(1-p_s\right) R_a^{SR} + 2q R_a^{RR} \right) \\
& + R_{az}^R + p_{single} \left( 2q\left(1-p_s\right) R_{az}^{SR} + 2q R_{az}^{RR} \right) \Big)/N
\end{aligned}
$$

$$
\begin{aligned}
\lambda_{XSR} = \Big( & \left(1-p_{single}\right)\left(\beta_S \theta_\beta p_s + \beta_S \theta_\beta f\left(1-p_s\right)\right) 2q \Big( S^{SR} + S_a^{SR} + E^{SR} + E_a^{SR} \\
& + I^{SR} + I_a^{SR} + I_{az}^{SR} + R^{SR} + R_a^{SR} + R_{az}^{SR} \Big)\Big)/N
\end{aligned}
$$

$$
\begin{aligned}
\lambda_{XSS} = \Big( & \left(1-p_{single}\right)\beta_S \theta_\beta 2q \big( S^{SS} + S_a^{SS} + E^{SS} + E_a^{SS} + I^{SS} + I_a^{SS} \\
& + I_{az}^{SS} + R^{SS} + R_a^{SS} + R_{az}^{SS} \big)\Big)/N
\end{aligned}
$$

$$
\begin{aligned}
\lambda_{XRR} = \Big( & \left(1-p_{single}\right)\beta_S \theta_\beta f 2q \big( S^{RR} + S_a^{RR} + E^{RR} + E_a^{RR} + I^{RR} + I_a^{RR} \\
& + I_{az}^{RR} + R^{RR} + R_a^{RR} + R_{az}^{RR} \big)\Big)/N
\end{aligned}
$$

SARS-CoV-2:

$$
\begin{aligned}
\lambda_C = \beta_C \theta_c \Big( & \big( I + I^S + I^R + I^{SR} + I^{SS} + I^{RR} + I_a + I_a^S + I_a^R + I_a^{SR} + I_a^{SS} + I_a^{RR} \\
& + I_{az} + I_{az}^S + I_{az}^R + I_{az}^{SR} + I_{az}^{SS} + I_{az}^{RR} \big)\Big)/N
\end{aligned}
$$

Clearance rates of pneumococcal carriage:

$$
\gamma^S = \gamma^R = \gamma^{SS} = \gamma^{RR} = \gamma^{SR}
$$

The equations for SARS-CoV-2 susceptible individuals (no antibiotic treatment):

$$
\begin{aligned}
\frac{dS}{dt} = \quad & -\left(\lambda_{XS} + \lambda_{XR} + \lambda_{XSR} + \lambda_{XSS} + \lambda_{XRR}\right) S + \gamma^S \left(S^S + S^{SS}\right) + \gamma^R \left(S^R + S^{SR} + S^{RR} + S_a^R + S_a^{RR}\right) \\
& + \left(\gamma^S + \omega\right)\left(S_a^S + S_a^{SS}\right) - \tau a S + r S_a - \lambda_C S
\end{aligned}
$$

$$
\frac{dS^S}{dt} = \lambda_{XS} S - \gamma^S S^S - k\lambda_S S^S - k\lambda_R S^S - \tau a S^S - \lambda_C S^S
$$

$$
\frac{dS^R}{dt} = \lambda_{XR} S - \gamma^R S^R - k\lambda_S S^R - k\lambda_R S^R - \tau a S^R + r S_a^R + \left(\gamma^S + \omega\right) S_a^{SR} - \lambda_C S^R
$$

$$
\begin{aligned}
\frac{dS^{SR}}{dt} = \quad & \lambda_{XSR} S - \gamma^R S^{SR} + k\lambda_R S^S + k\lambda_S S^R - kc\lambda_S S^{SR} - kc\lambda_R S^{SR} + 2kc \\
& \lambda_R S^{SS} + 2kc\lambda_S S^{RR} - \tau a S^{SR} - \lambda_C S^{SR}
\end{aligned}
$$

$$
\frac{dS^{SS}}{dt} = \lambda_{XSS} S - \gamma^S S^{SS} + k\lambda_S S^S + kc\lambda_S S^{SR} - 2kc\lambda_R S^{SS} - \tau a S^{SS} - \lambda_C S^{SS}
$$

$$
\frac{dS^{RR}}{dt} = \lambda_{XRR} S - \gamma^R S^{RR} + k\lambda_R S^R + kc\lambda_R S^{SR} - 2kc\lambda_S S^{RR} - \tau a S^{RR} + r S_a^{RR} - \lambda_C S^{RR}
$$

The equations for SARS-CoV-2 susceptible individuals (with antibiotic treatment):

$$\frac{dS_a}{dt} = -\left(\lambda_{XR} + \lambda_{XRR}\right) S_a + \tau a S - r S_a - \lambda_C S_a$$

$$\frac{dS_a^S}{dt} = -\left(\gamma^S + \omega\right) S_a^S + \tau a S^S - k\lambda_R S_a^S - \lambda_C S_a^S$$

$$\frac{dS_a^R}{dt} = -\gamma^R S_a^R + \lambda_{XR} S_a + \tau a S^R - r S_a^R - \lambda_C S_a^R$$

$$\frac{dS_a^{SR}}{dt} = -\left(\gamma^S + \omega\right) S_a^{SR} + k\lambda_R S_a^S + 2kc\lambda_R S_a^{SS} + \tau a S^{SR} - \lambda_C S_a^{SR}$$

$$\frac{dS_a^{SS}}{dt} = -\left(\gamma^S + \omega\right) S_a^{SS} - 2kc\lambda_R S_a^{SS} + \tau a S^{SS} - \lambda_C S_a^{SS}$$

$$\frac{dS_a^{RR}}{dt} = \lambda_{XRR} S_a + \tau a S^{RR} - r S_a^{RR} - \gamma^R S_a^{RR} - \lambda_C S_a^{RR}$$

The equations for SARS-CoV-2 exposed individuals (no antibiotic treatment):

$$\frac{dE}{dt} = -\left(\lambda_{XS} + \lambda_{XR} + \lambda_{XSR} + \lambda_{XSS} + \lambda_{XRR}\right) E + \gamma^S \left(E^S + E^{SS}\right) + \gamma^R \left(E^R + E^{SR} + E^{RR} + E_a^R + E_a^{RR}\right)$$
$$+ \left(\gamma^S + \omega\right) \left(E_a^S + E_a^{SS}\right) - \tau a E + r E_a + \lambda_C S - \varepsilon E$$

$$\frac{dE^S}{dt} = \lambda_{XS} E - \gamma^S E^S - k\lambda_S E^S - k\lambda_R E^S - \tau a E^S + \lambda_C S^S - \varepsilon E^S$$

$$\frac{dE^R}{dt} = \lambda_{XR} E - \gamma^R E^R - k\lambda_S E^R - k\lambda_R E^R - \tau a E^R + r E_a^R + \lambda_C S^R - \varepsilon E^R + \left(\gamma^S + \omega\right) E_a^{SR}$$

$$\frac{dE^{SR}}{dt} = \lambda_{XSR} E - \gamma^R E^{SR} + k\lambda_R E^S + k\lambda_S E^R - kc\lambda_S E^{SR} - kc\lambda_R E^{SR} + 2kc\lambda_R E^{SS} + 2kc\lambda_S E^{RR}$$
$$- \tau a E^{SR} + \lambda_C S^{SR} - \varepsilon E^{SR}$$

$$\frac{dE^{SS}}{dt} = \lambda_{XSS} E - \gamma^S E^{SS} + k\lambda_S E^S + kc\lambda_S E^{SR} - 2kc\lambda_R E^{SS} - \tau a E^{SS} + \lambda_C S^{SS} - \varepsilon E^{SS}$$

$$\frac{dE^{RR}}{dt} = \lambda_{XRR} E - \gamma^R E^{RR} + k\lambda_R E^R + kc\lambda_R E^{SR} - 2kc\lambda_S E^{RR} - \tau a E^{RR} + r E_a^{RR} + \lambda_C S^{RR} - \varepsilon E^{RR}$$

The equations for SARS-CoV-2 exposed individuals (with antibiotic treatment):

$$\frac{dE_a}{dt} = -\left(\lambda_{XR} + \lambda_{XRR}\right) E_a + \tau a E - r E_a + \lambda_C S_a - \varepsilon E_a$$

$$\frac{dE_a^S}{dt} = -\left(\gamma^S + \omega\right) E_a^S + \tau a E^S - k\lambda_R E_a^S + \lambda_C S_a^S - \varepsilon E_a^S$$

$$\frac{dE_a^R}{dt} = -\gamma^R E_a^R + \lambda_{XR} E_a + \tau a E^R - r E_a^R + \lambda_C E_a^R - \varepsilon E_a^R$$

$$\frac{dE_a^{SR}}{dt} = -\left(\gamma^S + \omega\right) E_a^{SR} + k\lambda_R E_a^S + 2kc\lambda_R E_a^{SS} + \tau a E^{SR} + \lambda_C S_a^{SR} - \varepsilon E_a^{SR}$$

$$\frac{dE_a^{SS}}{dt} = -\left(\gamma^S + \omega\right) E_a^{SS} - 2kc\lambda_R E_a^{SS} + \tau a E^{SS} + \lambda_C S_a^{SS} - \varepsilon E_a^{SS}$$

$$\frac{dE_a^{RR}}{dt} = \lambda_{XRR}E_a + \tau a E^{RR} - r E_a^{RR} - \gamma^R E_a^{RR} + \lambda_C S_a^{RR} - \varepsilon E_a^{RR}$$

The equations for SARS-CoV-2 infected individuals (no antibiotic treatment):

$$\begin{aligned}\frac{dI}{dt} = & -\left(\lambda_{XS} + \lambda_{XR} + \lambda_{XSR} + \lambda_{XSS} + \lambda_{XRR}\right) I + \gamma^S \left(I^S + I^{SS}\right) + \gamma^R \left(I^R + I^{SR} + I^{RR} + I_a^R + I_a^{RR}\right) \\ & + \left(\gamma^S + \omega\right)\left(I_a^S + I_a^{SS}\right) - \left(1 - p_{az}\right)\tau a I + r I_a + \varepsilon E - \gamma^C I - p_{az} I + \gamma^R \left(I_{az}^R + I_{az}^{RR}\right)\end{aligned}$$

$$\frac{dI^S}{dt} = \lambda_{XS} I - \gamma^S I^S - k\lambda_S I^S - k\lambda_R I^S - \left(1 - p_{az}\right)\tau a I^S + \varepsilon E^S - \gamma^C I^S - p_{az} I^S$$

$$\frac{dI^R}{dt} = \lambda_{XR} I - \gamma^R I^R - k\lambda_S I^R - k\lambda_R I^R + \left(\gamma^S + \omega\right) I_a^{SR} - \left(1 - p_{az}\right)\tau a I^R + r I_a^R + \varepsilon E^R - \gamma^C I^R - p_{az} I^R$$

$$\begin{aligned}\frac{dI^{SR}}{dt} = & \ \lambda_{XSR} I - \gamma^R I^{SR} + k\lambda_R I^S + k\lambda_S I^R - kc\lambda_S I^{SR} - kc\lambda_R I^{SR} + 2kc\lambda_R I^{SS} + 2kc\lambda_S I^{RR} \\ & - \left(1 - p_{az}\right)\tau a I^{SR} + \varepsilon E^{SR} - \gamma^C I^{SR} - p_{az} I^{SR}\end{aligned}$$

$$\frac{dI^{SS}}{dt} = \lambda_{XSS} I - \gamma^S I^{SS} + k\lambda_S I^S + kc\lambda_S I^{SR} - 2kc\lambda_R I^{SS} - \left(1 - p_{az}\right)\tau a I^{SS} + \varepsilon E^{SS} - \gamma^C I^{SS} - p_{az} I^{SS}$$

$$\frac{dI^{RR}}{dt} = \lambda_{XRR} I - \gamma^R I^{RR} + k\lambda_R I^R + kc\lambda_R I^{SR} - 2kc\lambda_S I^{RR} - \left(1 - p_{az}\right)\tau a I^{RR} + r I_a^{RR} + \varepsilon E^{RR} - \gamma^C I^{RR} - p_{az} I^{RR}$$

The equations for SARS-CoV-2 infected individuals (with antibiotic treatment):

$$\frac{dI_a}{dt} = -\left(\lambda_{XR} + \lambda_{XRR}\right) I_a + \left(1 - p_{az}\right)\tau a I - r I_a + \varepsilon E_a - \gamma^C I_a$$

$$\frac{dI_a^S}{dt} = -\left(\gamma^S + \omega\right) I_a^S + \left(1 - p_{az}\right)\tau a I^S - k\lambda_R I_a^S + \varepsilon E_a^S - \gamma^C I_a^S$$

$$\frac{dI_a^R}{dt} = -\gamma^R I_a^R + \lambda_{XR} I_a + \left(1 - p_{az}\right)\tau a I^R - r I_a^R + \varepsilon E_a^R - \gamma^C I_a^R$$

$$\frac{dI_a^{SR}}{dt} = -\left(\gamma^S + \omega\right) I_a^{SR} + k\lambda_R I_a^S + 2kc\lambda_R I_a^{SS} + \left(1 - p_{az}\right)\tau a I^{SR} + \varepsilon E_a^{SR} - \gamma^C I_a^{SR}$$

$$\frac{dI_a^{SS}}{dt} = -\left(\gamma^S + \omega\right) I_a^{SS} - 2kc\lambda_R I_a^{SS} + \left(1 - p_{az}\right)\tau a I^{SS} + \varepsilon E_a^{SS} - \gamma^C I_a^{SS}$$

$$\frac{dI_a^{RR}}{dt} = \lambda_{XRR} I_a + \left(1 - p_{az}\right)\tau a I^{RR} - r I_a^{RR} - \gamma^R I_a^{RR} + \varepsilon E_a^{RR} - \gamma^C I_a^{RR}$$

The equations for SARS-CoV-2 infected individuals (with azithromycin treatment):

$$\frac{dI_{az}}{dt} = p_{az} I - \gamma^C I_{az} - \left(\lambda_{XR} + \lambda_{XRR}\right) I_{az} + \left(\gamma^S + \omega\right)\left(I_{az}^S + I_{az}^{SS}\right)$$

$$\frac{dI_{az}^S}{dt} = p_{az} I^S - \gamma^C I_{az}^S - \left(\gamma^S + \omega\right) I_{az}^S - k\lambda_R I_{az}^S$$

$$\frac{dI_{az}^R}{dt} = p_{az} I^R - \gamma^C I_{az}^R + \lambda_{XR} I_{az} - \gamma^R I_{az}^R + \left(\gamma^S + \omega\right) I_{az}^{SR}$$

$$\frac{dI_{az}^{SR}}{dt} = p_{az} I^{SR} - \gamma^C I_{az}^{SR} - \left(\gamma^S + \omega\right) I_{az}^{SR} + k\lambda_R I_{az}^S + 2kc\lambda_R I_{az}^{SS}$$

$$\frac{dI_{az}^{SS}}{dt} = p_{az}I^{SS} - \gamma^C I_{az}^{SS} - \left(\gamma^S + \omega\right) I_{az}^{SS} - 2kc\lambda_R I_{az}^{SS}$$

$$\frac{dI_{az}^{RR}}{dt} = p_{az}I^{RR} - \gamma^C I_{az}^{RR} + \lambda_{XRR}I_{az} - \gamma^R I_{az}^{RR}$$

The equations for SARS-CoV-2 recovered individuals (no antibiotic treatment):

$$\begin{aligned}
\frac{dR}{dt} = \ & -\left(\lambda_{XS} + \lambda_{XR} + \lambda_{XSR} + \lambda_{XSS} + \lambda_{XRR}\right) R + \gamma^S \left(R^S + R^{SS}\right) \\
& + \gamma^R \left(R^R + R^{SR} + R^{RR} + R_a^R + R_a^{RR}\right) + \left(\gamma^S + \omega\right) \left(R_a^S + R_a^{SS}\right) - \tau a R + r R_a \\
& + \gamma^C I + r_{az}R_{az} + \gamma^R \left(R_{az}^R + R_{az}^{RR}\right)
\end{aligned}$$

$$\frac{dR^S}{dt} = \lambda_{XS}R - \gamma^S R^S - k\lambda_S R^S - k\lambda_R R^S - \tau a R^S + \gamma^C I^S + r_{az}R_{az}^S$$

$$\frac{dR^R}{dt} = \lambda_{XR}R - \gamma^R R^R - k\lambda_S R^R - k\lambda_R R^R + \left(\gamma^S + \omega\right) R_a^{SR} - \tau a R^R + r R_a^R + \gamma^C I^R + r_{az}R_{az}^R$$

$$\begin{aligned}
\frac{dR^{SR}}{dt} = \ & \lambda_{XSR}R - \gamma^R R^{SR} + k\lambda_R R^S + k\lambda_S R^R - kc\lambda_S R^{SR} - kc\lambda_R R^{SR} + 2kc\lambda_R R^{SS} + 2kc\lambda_S R^{RR} \\
& - \tau a R^{SR} + \gamma^C I^{SR} + r_{az}R_{az}^{SR}
\end{aligned}$$

$$\frac{dR^{SS}}{dt} = \lambda_{XSS}R - \gamma^S R^{SS} + k\lambda_S R^S + kc\lambda_S R^{SR} - 2kc\lambda_R R^{SS} - \tau a R^{SS} + \gamma^C I^{SS} + r_{az}R_{az}^{SS}$$

$$\frac{dR^{RR}}{dt} = \lambda_{XRR}R - \gamma^R R^{RR} + k\lambda_R R^R + kc\lambda_R R^{SR} - 2kc\lambda_S R^{RR} - \tau a R^{RR} + r R_a^{RR} + \gamma^C I^{RR} + r_{az}R_{az}^{RR}$$

The equations for SARS-CoV-2 recovered individuals (with antibiotic treatment):

$$\frac{dR_a}{dt} = \gamma^C I_a + \tau a R - r R_a - \lambda_{XR}R_a - \lambda_{XRR}R_a$$

$$\frac{dR_a^S}{dt} = \gamma^C I_a^S + \tau a R^S - \left(\gamma^S + \omega\right) R_a^S - k\lambda_R R_a^S$$

$$\frac{dR_a^R}{dt} = \gamma^C I_a^R + \tau a R^R - r R_a^R + \lambda_{XR}R_a - \gamma^R R_a^R$$

$$\frac{dR_a^{SR}}{dt} = \gamma^C I_a^{SR} + \tau a R^{SR} - \left(\gamma^S + \omega\right) R_a^{SR} + k\lambda_R R_a^S + 2kc\lambda_R R_a^{SS}$$

$$\frac{dR_a^{SS}}{dt} = \gamma^C I_a^{SS} + \tau a R^{SS} - \left(\gamma^S + \omega\right) R_a^{SS} - 2kc\lambda_R R_a^{SS}$$

$$\frac{dR_a^{RR}}{dt} = \gamma^C I_a^{RR} + \tau a R^{RR} - r R_a^{RR} + \lambda_{XRR}R_a - \gamma^R R_a^{RR}$$

The equations for SARS-CoV-2 recovered individuals (with azithromycin treatment):

$$\frac{dR_{az}}{dt} = \gamma^C I_{az} - r_{az}R_{az} - \lambda_{XR}R_{az} - \lambda_{XRR}R_{az} + \left(\gamma^S + \omega\right) \left(R_{az}^S + R_{az}^{SS}\right)$$

$$\frac{dR_{az}^S}{dt} = \gamma^C I_{az}^S - r_{az}R_{az}^S - \left(\gamma^S + \omega\right) R_{az}^S - k\lambda_R R_{az}^S$$

$$\frac{dR_{az}^R}{dt} = \gamma^C I_{az}^R - r_{az}R_{az}^R + \lambda_{XR}R_{az} - \gamma^R R_{az}^R + \left(\gamma^S + \omega\right) R_{az}^{SR}$$

$$\frac{dR_{az}^{SR}}{dt} = \gamma^C I_{az}^{SR} - r_{az}R_{az}^{SR} - \left(\gamma^S + \omega\right) R_{az}^{SR} + k\lambda_R R_{az}^S + 2kc\lambda_R R_{az}^{SS}$$

$$\frac{dR_{az}^{SS}}{dt} = \gamma^C I_{az}^{SS} - r_{az}R_{az}^{SS} - \left(\gamma^S + \omega\right) R_{az}^{SS} - 2kc\lambda_R R_{az}^{SS}$$

$$\frac{dR_{az}^{RR}}{dt} = \gamma^C I_{az}^{RR} - r_{az}R_{az}^{RR} + \tau a R^{RR} - r R_a^{RR} + \lambda_{XRR}R_a - \gamma^R R_a^{RR}$$

The equations for invasive pneumococcal disease (IPD) incidence caused by sensitive and resistant strains:

$$\frac{dIPD^S}{dt} = p_{inv}IPD_{risk}\left(S^S + S^{SS} + \frac{1}{2}S^{SR} + E^S + E^{SS} + \frac{1}{2}E^{SR} + I^S + I^{SS} + \frac{1}{2}I^{SR} + R^S + R^{SS} + \frac{1}{2}R^{SR}\right)$$

$$\frac{dIPD^R}{dt} = p_{inv}IPD_{risk}\Big(S^R + S^{RR} + \frac{1}{2}S^{SR} + E^R + E^{RR} + \frac{1}{2}E^{SR} + I^R + I^{RR} + \frac{1}{2}I^{SR} + R^R + R^{RR}$$
$$+\frac{1}{2}R^{SR} + S_a^R + S_a^{RR} + S_a^{SR} + E_a^R + E_a^{RR} + E_a^{SR} + I_a^R + I_a^{RR} + I_a^{SR}$$
$$+R_a^R + R_a^{RR} + R_a^{SR} + I_{az}^R + I_{az}^{RR} + I_{az}^{SR} + R_{az}^R + R_{az}^{RR} + R_{az}^{SR}\Big)$$

## Equations for the prevalence of pneumococcal carriage in the population:

Prevalence of antibiotic-sensitive pneumococcal carriage is calculated as:

$$S_{prop} = \Big(S^S + \frac{1}{2}S^{SR} + S^{SS} + S_a^S + \frac{1}{2}S_a^{SR} + S_a^{SS} + E^S +$$
$$\frac{1}{2}E^{SR} + E^{SS} + E_a^S + \frac{1}{2}E_a^{SR} + E_a^{SS} + I^S + \frac{1}{2}I^{SR} + I^{SS} + I_a^S +$$
$$\frac{1}{2}I_a^{SR} + I_a^{SS} + I_{az}^S + \frac{1}{2}I_{az}^{SR} + I_{az}^{SS} + R^S + \frac{1}{2}R^{SR} + R^{SS} + R_a^S +$$
$$\frac{1}{2}R_a^{SR} + R_a^{SS} + R_{az}^S + \frac{1}{2}R_{az}^{SR} + R_{az}^{SS}\Big)/Total\ carriage$$

Prevalence of antibiotic-resistant pneumococcal carriage is calculated as:

$$R_{prop} = \Big(S^R + \frac{1}{2}S^{SR} + S^{RR} + S_a^R + \frac{1}{2}S_a^{SR} + S_a^{RR} + E^R +$$
$$\frac{1}{2}E^{SR} + E^{RR} + E_a^R + \frac{1}{2}E_a^{SR} + E_a^{RR} + I^R + \frac{1}{2}I^{SR} + I^{RR} + I_a^R +$$
$$\frac{1}{2}I_a^{SR} + I_a^{RR} + I_{az}^R + \frac{1}{2}I_{az}^{SR} + I_{az}^{RR} + R^R + \frac{1}{2}R^{SR} + R^{RR} + R_a^R +$$
$$\frac{1}{2}R_a^{SR} + R_a^{RR} + R_{az}^R + \frac{1}{2}R_{az}^{SR} + R_{az}^{RR}\Big)/Total\ carriage$$

Total pneumococcal carriage is calculated as:

$$Total\ carriage = S_{prop} + R_{prop}$$

Prevalence of total pneumococcal carriage is calculated as:

$$Total_{prop} = Total\ carriage/N$$

## Equations for within-host interactions:

Invasive pneumococcal disease (IPD) incidence implementing within-host interactions is only accounted for in the equations for IPD incidence outside of the model equations, and hence has no impact on model dynamics.

Daily IPD incidence, for the antibiotic-resistant strain is calculated as:

$$\frac{dIPD^R}{dt} = p_{inv}IPD_{risk}\left(S^R + S^{RR} + \frac{1}{2}S^{SR} + E^R + E^{RR} + \frac{1}{2}E^{SR} + \right.$$
$$\psi_c\left(I^R + I^{RR} + \frac{1}{2}I^{SR}\right) + R^R + R^{RR} + \frac{1}{2}R^{SR} + S_a^R +$$
$$S_a^{RR} + S_a^{SR} + E_a^R + E_a^{RR} + E_a^{SR} +$$
$$\psi_c\left(I_a^R + I_a^{RR} + I_a^{SR}\right) + R_a^R + R_a^{RR} + R_a^{SR} +$$
$$\left.\psi_c\left(I_{az}^R + I_{az}^{RR} + I_{az}^{SR}\right) + R_{az}^R + R_{az}^{RR} + R_{az}^{SR}\right)$$

where $p_{inv}$ represents the bacterial invasion rate or progression of carriage to invasive disease and $\psi_c$ represents the co-infection interaction term that increases the rate of progression to invasive disease among colonized individuals who are also infected with SARS-CoV-2.

Daily IPD incidence for the antibiotic-sensitive strain is calculated as:

$$\frac{dIPD^S}{dt} = p_{inv}IPD_{risk}\left(S^R + S^{RR} + \frac{1}{2}S^{SR} + E^R + E^{RR} + \frac{1}{2}E^{SR}\right.$$
$$+\psi_c\left(I^R + I^{RR} + \frac{1}{2}I^{SR}\right) + R^R + R^{RR} + \frac{1}{2}R^{SR} + S_a^R$$
$$+S_a^{RR} + S_a^{SR} + E_a^R + E_a^{RR} + E_a^{SR}$$
$$+\psi_c\left(I_a^R + I_a^{RR} + I_a^{SR}\right) + R_a^R + R_a^{RR} + R_a^{SR}$$
$$\left.+\psi_c\left(I_{az}^R + I_{az}^{RR} + I_{az}^{SR}\right) + R_{az}^R + R_{az}^{RR} + R_{az}^{SR}\right)$$

Due to large uncertainty in the values of $\psi_c$ we considered broad range in our analysis estimated from prior modelling studies (*Domenech de Cellès et al., 2019*; *Opatowski et al., 2013*). In the context of ecological interactions between *S. pneumoniae* and influenza, estimations show that influenza co-infection increases the rate of progression from *S. pneumonia* colonization to invasive pneumococcal disease between 49 and 146-fold, on average, depending on the age group (*Domenech de Cellès et al., 2019*; *Opatowski et al., 2013*). Therefore, we implemented the range $1 < \psi_c < 100$ for SARS-CoV-2.

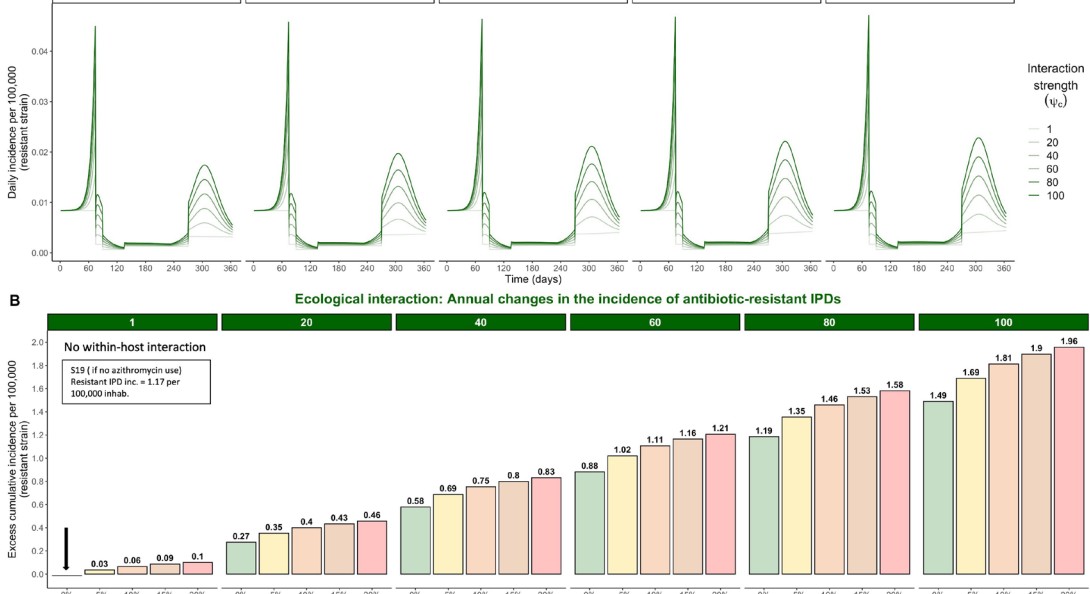

**Appendix 2—figure 1.** Hypothetical within-host interactions in scenario S19 promote the incidence of antibiotic-resistant invasive pneumococcal disease (IPD). (**A**) Impacts of ecological interactions between pathogens. When SARS-CoV-2 infection leads to faster progression from pneumococcal colonization to disease , surges in COVID-19 cases accompanied by increasing levels of azithromycin use lead to substantial increases in the daily incidence of

*Appendix 2—figure 1 continued on next page*

*Appendix 2—figure 1 continued*

antibiotic-resistant IPD. (**B**) Annual excess in cumulative IPD incidence due to synergistic within-host ecological interactions. A rate of disease progression increased by a factor (no within-host interaction) and in scenario S19 applied to the general population resulted in approximately 0.06 and 0.75 additional cases of antibiotic-resistant disease per 100,000 inhabitants over the course of one year, respectively, compared to the scenario S19 assuming no within-host interaction and no azithromycin use indicated by the black arrow (1.17 cases/100,000 inhabitants).

### A  SARS-CoV-2 transmission and infection:

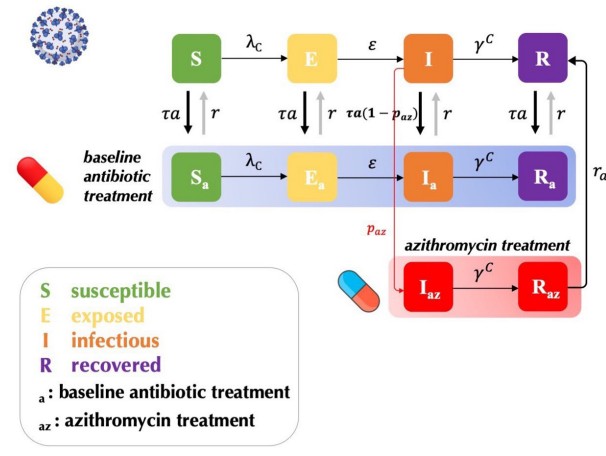

$\lambda_C$ = force of infection

$\varepsilon$ = 1/duration of the latent period

$\gamma^C$ = SARS-CoV-2 recovery rate

$\tau$ = baseline rate of antibiotic prescribing in the community, i.e., daily nb. of prescriptions per capita (ppc)

$a$ = indirect effects of the pandemic on the community antibiotic prescribing (decrease)

$r$ = 1/avg. duration of the baseline antibiotic treatment

$r_{az}$ = 1/remaining exposure to azithromycin after the initial 7 days (1/11.5days)

$p_{az}$ = a proportion of the SARS-CoV-2 infected in the community (or in a particular age group within the community) using azithromycin (0-0.2)

**S**  susceptible
**E**  exposed
**I**  infectious
**R**  recovered
**a**: baseline antibiotic treatment
**az**: azithromycin treatment

### B  Pneumococcal carriage transmission and progression to invasive pneumococcal disease within each of the SEIR model compartments:

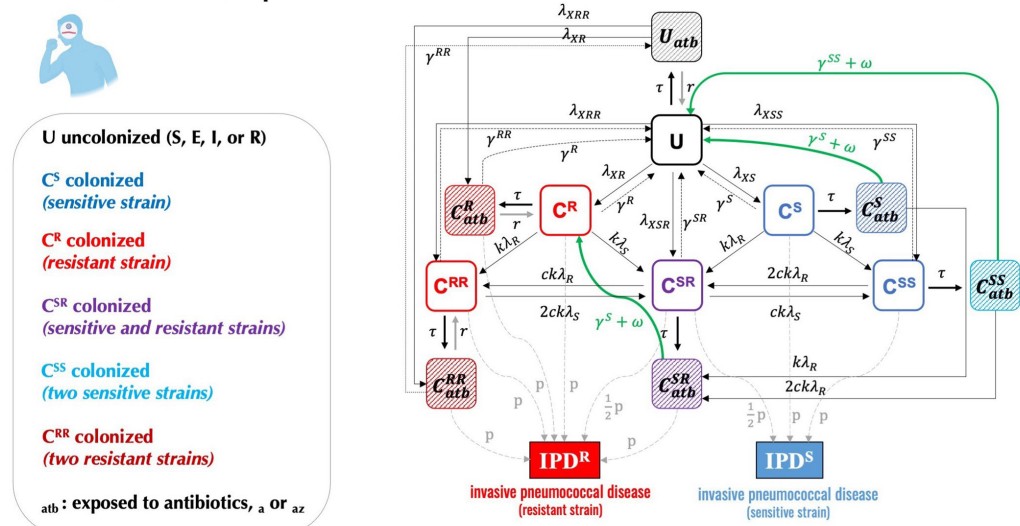

**U** uncolonized (S, E, I, or R)

**C$^S$** colonized
*(sensitive strain)*

**C$^R$** colonized
*(resistant strain)*

**C$^{SR}$** colonized
*(sensitive and resistant strains)*

**C$^{SS}$** colonized
*(two sensitive strains)*

**C$^{RR}$** colonized
*(two resistant strains)*

**atb**: exposed to antibiotics, **a** or **az**

**Appendix 2—figure 2.** Model schematic describing co-circulation of SARS-CoV-2 infection and pneumococcal carriage transmission. (**A**) Individuals in each of the SARS-CoV-2 transmission model compartments (S, E, I, R) can be either (**B**) uncolonized (U) or colonized by the bacteria (C$^S$, C$^R$, C$^{SS}$, C$^{RR}$, C$^{SR}$) with or without concomitant antibiotic treatment (subscript $_{atb}$ denotes compartments exposed to antibiotics – both standard baseline antibiotic exposure and azithromycin exposure). Full list of model parameters can be found in *Appendix 2—table 2*.

**Appendix 2—table 1.** Total number of invasive *S. pneumoniae* isolates from blood or cerebrospinal fluid tested in 2019 and 2020 and reported to EARS-Net (European Antimicrobial Resistance Surveillance Network).

| Country | No. of invasive Streptococcus pneumoniae isolates in the EU/EEA | | % decrease in the total number of reported invasive isolates from 2019–2020 |
|---------|------|------|------|
| | 2019 | 2020 | |
| Austria | 550 | 301 | 45.3% |
| Belgium | 1548 | 858 | 44.6% |
| Bulgaria | 46 | 28 | 39.1% |
| Croatia | 156 | 55 | 64.7% |
| Czechia | 387 | 204 | 47.3% |
| Denmark | 601 | 351 | 41.6% |
| Estonia | 161 | 80 | 50.3% |
| Finland | 678 | 293 | 56.8% |
| France | 1264 | 668 | 47.2% |
| Germany | 2035 | 1314 | 35.4% |
| Hungary | 222 | 124 | 44.1% |
| Iceland | 44 | 20 | 54.5% |
| Ireland | 348 | 136 | 60.9% |
| Italy | 1351 | 685 | 49.3% |
| Latvia | 79 | 42 | 46.8% |
| Lithuania | 120 | 96 | 20.0% |
| Luxembourg | 38 | 24 | 36.8% |
| Malta | 27 | 16 | 40.7% |
| Netherlands | 1552 | 997 | 35.8% |
| Norway | 507 | 243 | 52.1% |
| Poland | 364 | 165 | 54.7% |
| Portugal | 983 | 588 | 40.2% |
| Romania | 107 | 42 | 60.7% |
| Slovakia | 40 | 15 | 62.5% |
| Slovenia | 283 | 172 | 39.2% |
| Spain | 1038 | 611 | 41.1% |
| Sweden | 1071 | 551 | 48.6% |
| United Kingdom | 3468 | 1412 | 59.3% |

**Appendix 2—table 2.** Summary of the model parameters.

| Symbol | Interpretation | Value(s) | References |
|--------|----------------|----------|------------|
| SARS-CoV-2 infection parameters: | | | |
| $\beta_C$ | transmission rate of SARS-CoV-2 | 0.46 days$^{-1}$ | *Liu et al., 2020* |
| $\gamma^c$ | SARS-CoV-2 recovery rate (mild cases) | 1/7 days$^{-1}$ | *Lauer et al., 2020*; *Rhee et al., 2020* |

*Appendix 2—table 2 Continued on next page*

*Appendix 2—table 2 Continued*

| Symbol | Interpretation | Value(s) | References |
|---|---|---|---|
| $\epsilon$ | SARS-CoV-2 incubation rate (latent period from exposed to infectious state) | 1/5 days$^{-1}$ | *Elias et al., 2021* |
| $\theta_c$ | Relative risk of SARS-CoV-2 transmission due to lockdown implementation | 0.23 | *Salje et al., 2020* |
| Pneumococcal colonization and invasion parameters: | | | |
| $\beta_s$ | transmission rate of antibiotic-sensitive strain | 0.056 days$^{-1}$<br>0.046 days$^{-1}$<br>0.034 days$^{-1}$ | *Davies et al., 2019*;<br>*Olesen et al., 2020* |
| $f$ | fitness of antibiotic-resistant strain (assuming there is a fitness cost on transmissibility) | 0.9652<br>0.949<br>0.926 | *Dagan et al., 2008*;<br>*Melnyk et al., 2015* |
| $\beta_s f$ | transmission rate of antibiotic-resistant strain | | calculated |
| $\theta_\beta$ | Relative risk of pneumococcal transmission due to lockdown implementation | 1 or 0.75 | assumed |
| $\gamma^S = \gamma^R = \gamma^{SS}$<br>$= \gamma^{RR} = \gamma^{SR}$ | rate of natural bacterial clearance (assumed to be the same for antibiotic-sensitive and -resistant strains) | 1/20 days$^{-1}$<br>1/30 days$^{-1}$<br>1/45 days$^{-1}$ | *Abdullahi et al., 2012*; *Davies et al., 2019*; *Ekdahl et al., 1997*; *Högberg et al., 2021* *Melegaro et al., 2004* |
| $q$ | relative infectiousness with each strain for dually colonized | 0.5 | assumed *Colijn et al., 2010* |
| $c$ | fraction of dually colonized returning to single-colonized upon reinfection | 0.5 | assumed (*Colijn et al., 2010*) |
| $r$ | probability of acquiring secondary bacterial carriage | 0.5 | assumed (*Colijn et al., 2010*) |
| $p_s$ | probability of transmitting antibiotic-sensitive strain | 0.5 | assumed (*Colijn et al., 2010*) |
| $p_{single}$ | probability of a single infection | 0.5 | assumed (*Colijn et al., 2010*) |
| $p_{inv}$ | pneumococcal invasion rate (summer and winter) | [3x10$^{-6}$ day$^{-1}$,9x10$^{-6}$ day$^{-1}$] in the elderly and general population, and [1x10$^{-6}$ day$^{-1}$, 2.5x10$^{-6}$ day$^{-1}$] in <5 years-old | *Domenech de Cellès et al., 2019*; *Opatowski et al., 2013* |
| $S^S + S^R$ | initial states – initial prevalence of the total pneumococcal carriage (antibiotic-sensitive and -resistant) in different populations | 10%<br>20%<br>30% | *Cohen et al., 2023*; *Rose et al., 2021*; *Rybak et al., 2022*; *Tinggaard et al., 2023*; *Wang et al., 2017* |
| Antibiotic exposure parameters: | | | |
| $\omega$ | rate of antibiotic-induced pneumococcal clearance for sensitive strains(1/time before antibiotic action) | 1/3 days$^{-1}$ | *Kuitunen et al., 2023* |

*Appendix 2—table 2 Continued on next page*

*Appendix 2—table 2 Continued*

| Symbol | Interpretation | Value(s) | References |
|---|---|---|---|
| $r$ | rate of return to antibiotic unexposed compartment (1/duration of antibiotic treatment) | 1/7 days$^{-1}$ | *Grant and Saux, 2021*; *Kuitunen et al., 2023* |
| $r_{az}$ | rate of return to antibiotic unexposed compartment (1/the remainder of how long azithromycin stays in the body) | 1/11.5 days$^{-1}$ | calculated (*Foulds et al., 1990*; *Girard et al., 2005*) |
| $\tau$ | baseline rate of antibiotic exposure in the community (France) | 0.0014 average daily ppc (prescriptions per capita) | *Bara et al., 2022* |
| $a$ | A reduction factor for antibiotic exposure in the community resulting from changes in healthcare-seeking behavior in response to the COVID-19 pandemic | [0.51, 0.77, 0.84] to represent annual 13%, 18%, and 39% decrease observed in France | *Bara et al., 2022* |
| $p_{az}$ | A proportion of COVID-19 infected individuals in the community receiving azithromycin | [0–0.20] testing between 0% and 20% | *Tsay et al., 2022*; *Wittman et al., 2023* |
| Pathogenicity (invasive pneumococcal disease risk): | | | |
| $IPD_{risk}$ | A reduction factor for the risk of developing an invasive pneumococcal disease (IPD) due to the absence of influenza-like-illnesses (ILIs) after lockdown implementation | 1 (pre-lockdown) 0.2 (lockdown) 0.4 (post-lockdown) for an average of 0.5 in 2020 | *Shaw et al., 2023* |

