## [Editor Report]

The mathematical modeling approach disentangles the impacts of the COVID-19 pandemic on antibiotic resistance in Streptococcus pneumoniae in the community setting and identifies the three most plausible driving mechanisms responsible for the observed trends in invasive isolates and pneumococcal carriage. The significance of the findings is important for our understanding of the changed epidemiology of invasive pneumococcal infections during the COVID19 pandemic. The strength of the evidence is graded as solid, derived from a well-described mathematical model evaluating multiple scenarios.

---

## [Decision Letter]

**Decision letter after peer review:**

Thank you for sending your article entitled "A modeling study on the impact of COVID-19 pandemic responses on the community transmission of antibiotic-resistant bacteria" for peer review at *eLife*. Your article is being evaluated by 2 peer reviewers, and the evaluation is being overseen by a Reviewing Editor and Jos van der Meer as the Senior Editor.

Both reviewers raised concerns about the design of the model (it not being non-neutral) and the assumptions made for antibiotic use in the community.

*Reviewer #1 (Recommendations for the authors):*

The authors have developed a mathematical model to investigate the impact of the SARS-CoV-2 pandemic on the spread of antibiotic-susceptible and antibiotic-resistant strains. The strength is the proof of concept, which shows the direction of incidence and resistance levels certain scenarios have, I am less convinced by the exact numerical values.

I was wondering about the model structure, which is not a neutral null model. For instance, why is superinfection possible with two serotypes if one is susceptible and one is resistant, but not if they are both susceptible or resistant?

As a result of the non-neutrality, the strain with the lowest prevalence has a selective advantage over the other strain as the pool of individuals they can infect is larger for the low-prevalent strain. Moreover, if c=0.05, in absence of AB use, about 10% of individuals will carry resistant strains, despite c>0. Hence there is a hidden mechanism that promotes the strain with the lowest prevalence. I would prefer to see a neutral null model, although I expect that the directions on incidence and resistance levels will not change a lot.

I do not understand the parameter choice in Figure 3. According to Figure 3, over 50% of the strains are resistant. However, in Figure 1 no country had a pre-pandemic resistance proportion for penicillin or macrolide resistance of 50% or more.

Details, I would not use the word rate for the proportion/fraction resistance. For me, the rate is an incidence per time unit.

In line 263, I think mentioning of a relative increase may be more intuitively understandable than an absolute increase.

*Reviewer #2 (Recommendations for the authors):*

In this manuscript the authors propose a mathematical model to explore the effects of the COVID-19 pandemic on the epidemiology of S. pneumoniae, considering both direct effects of SARS-CoV-2 infection, which is assumed to be capable of increasing the rate of progression from colonization with S. pneumoniae to invasive diseases, and indirect effects. The indirect effects are assumed to be the result of (i) changes in contact patterns between individuals (eg through population-wide "lockdowns" and isolation of COVID-19 cases), and (ii) changes in antibiotic consumption which is assumed to be the result of both widespread antibiotic prophylaxis amongst those infected with SARS-CoV-2 and changes in healthcare seeking behaviour. The questions addressed are of widespread interest, and the authors are to be congratulated for a first attempt at addressing this difficult problem. Strengths of the work include a well-motivated research question, a largely clear exposition of the model and findings, and a generally well-written manuscript. I have no concerns about the implementation of the models. I do, however, have a number of concerns about the model assumptions that lead me to question whether the observed changes in the incidence of invasive disease with S. pneumoniae and in the prevalence of resistance amongst invasive isolates are really driven by the mechanisms encapsulated by the proposed models. I expand on these below.

One concern is that this study focuses on the community transmission of bacteria, and (unless I have fundamentally misunderstood the work) a central assumption seems to be that high rates of antibiotics were used in the community as prophylaxis amongst patients with COVID-19. As S2.2 explains, in the scenarios considered 18.1%, 33.0%, and 55.1% of patients received antibiotic prophylaxis during each COVID infection. The authors state "As reported in Langford et al. [3], a meta-analysis on antibiotic prescribing in patients with COVID-19, prophylactic antibiotic prescribing in the inpatient/outpatient setting was estimated at 59.3%. Therefore, the values above that we use in our model are conservative compared to this estimate." There are two issues here: first I couldn't find where the number of 59.3% in the Langford et al. paper came from (searching for 59 in the manuscript gave no relevant hits). In fact, this paper reports even higher rates of antibiotic use "The majority of patients with COVID-19 received antibiotics (71.9%, 95%CI 56.1 to 87.7%)." (elsewhere in the paper also reported as 71.8%). The second (and more important) issue is that even though the scope of Langford et al. included hospital and community settings, in all 24 of the included studies the setting was either ICU (n=5) or hospital (n=19). There is no reason to think these numbers are at all representative of the community setting (the focus of the manuscript) and the assumption that in 18% of COVID-19 infections (including, presumably, subclinical infections) antibiotics were taken prophylactically in the community does not seem credible to me (indeed, from personal experience, I am not aware of a single person who took antibiotics prophylactically in the community while infected with COVID). This might well just reflect my ignorance – perhaps there were countries where antibiotic prophylaxis in the community was widespread, but this has not been shown in the paper by Langford et al., and given that this is a central premise of the model and something that is presumably driving a lot of the results this is surely something where the data supporting this assumption needs to be made very clear if indeed such data exists. If, as I suspect is true in most settings, antibiotic prophylaxis during COVID infection in the community was negligible then I do think there is a need to make appropriate changes to the manuscript.

A second concern is in the treatment of the progression from colonisation to invasive disease. The authors assume that infection with SARS-CoV-2 increases the rate of progression (and consider up to 100-fold increases). This itself is reasonable. However, as the authors acknowledge, other respiratory viral infections also have this effect (SI-8: "estimations show that influenza co-infection increases the rate of progression from S. pneumonia colonization to invasive pneumococcal disease between 49 and 146-fold, on average, depending on the age group"). Since a major effect of the changes in contact patterns in response to the pandemic was that in many countries other respiratory infections, including influenza, did not circulate as usual, wouldn't this be expected to also have a substantial impact on the rate of progression from colonization to invasive disease? i.e. if it is important to consider the impact of SARS-CoV-2 on the progression, why not also consider the effect of other viral pathogens at the same time? A similar concern applies to antibiotic use. We know that in interpandemic periods a lot of antibiotic use in the community is given in response to viral respiratory infections, and since we know that changes in contact patterns had a profound impact on many such viral respiratory infections, is it not also likely that these had an effect on antibiotic use? Or is the assumption that such changes are just absorbed into changes in health-seeking behaviour?

A third concern is really a question about whether there might be another mechanism that could be explaining the generally increasing trends in penicillin and macrolide resistance in invasive pneumococcal disease summarised in figure 1. We know that the S. pneumoniae serotypes which have a longer carriage duration are more likely to be resistant (see, for example, Lehtinen et al., PNAS 2017), and also that these have a higher reproduction number (the shorter duration of carriage serotypes being maintained by negative frequency-dependent selection). We would therefore surely expect these less resistant serotypes (with lower reproduction numbers and shorter durations of carriage) to be more impacted by social distancing measures as they would be the first serotypes to have reproduction numbers reduced to below one. This would then be a completely different mechanism to explain observed changes in resistance patterns. Clearly, a lot of work would be needed to establish whether or not this can explain observed changes, though I feel the same is true of the explanatory mechanisms proposed in this paper (increased antibiotic use) given that the current paper lacks any country-specific antibiotic use data or attempt to see if different changes in patterns of antibiotic use can explain the different changes in resistance seen in different countries.

1. lines 118-120 highlight azithromycin as a particular concern for overuse, but this is difficult to square with the review the authors cite from Langford which reports macrolides as accounting for only 6.5% of antibiotic prescribing in COVID patients. This review suggests fluoroquinolones and β-lactams accounted for over 80% of prescribed antibiotics

2. On first looking at the manuscript I was quite concerned that the changes in resistance in figure 1 might reflect changes in health-seeking behaviour rather than real changes in the underlying epidemiology, for example, if certain types of isolates are less likely to be resistant and these types were reported less frequently. Now that I realise that the data in figure 1 all come from invasive isolates (as stated on line 176) this explanation seems less likely. It would be helpful if figure 1 could be labelled appropriately to make it clear that the data shown represents only invasive isolates.

3. line 248 "some respiratory viruses are known to favor bacterial disease" I don't think "favor" is the right word here. Perhaps "increase the risks of"?

4. Figure 3 would be easier to read if the subgraphs were labelled with the scenario. Also, it wasn't clear to me what the three graphs in panel C represent.

5. Table S3. I think A=20 in all the scenarios so that a minimum of 18% of patients infected with SARS-CoV-2 are given antibiotics prophylactically. It would increase readability if this was made explicit in the table.

---

## [Author Response]

Reviewer #1 (Recommendations for the authors):The authors have developed a mathematical model to investigate the impact of the SARS-CoV-2 pandemic on the spread of antibiotic-susceptible and antibiotic-resistant strains. The strength is the proof of concept, which shows the direction of incidence and resistance levels certain scenarios have, I am less convinced by the exact numerical values.I was wondering about the model structure, which is not a neutral null model. For instance, why is superinfection possible with two serotypes if one is susceptible and one is resistant, but not if they are both susceptible or resistant?As a result of the non-neutrality, the strain with the lowest prevalence has a selective advantage over the other strain as the pool of individuals they can infect is larger for the low-prevalent strain. Moreover, if c=0.05, in absence of AB use, about 10% of individuals will carry resistant strains, despite c>0. Hence there is a hidden mechanism that promotes the strain with the lowest prevalence. I would prefer to see a neutral null model, although I expect that the directions on incidence and resistance levels will not change a lot.

Thank you for your comment, we indeed used the initial model structure for simplicity. However, we acknowledge the importance of model neutrality. In this new version, we modified the component of the model pertaining to pneumococcal carriage, encompassing all three dually infected states, and accounted for the simultaneous transmission of a dual infection, such as in Colijn et al., (J R Soc Interface 2010) and updated equations for the forces of infection accordingly (Appendix 2). These changes did not affect the model’s results.

**Author response image 1. sa2fig1:** 

I do not understand the parameter choice in Figure 3. According to Figure 3, over 50% of the strains are resistant. However, in Figure 1 no country had a pre-pandemic resistance proportion for penicillin or macrolide resistance of 50% or more.

We thank the reviewer for raising that point. Our model initialization was overestimating the observed resistance rate in France for invasive isolates in 2019. The previous version of the model initially only aimed at tracking trends instead of exact numbers. This new version is much more closely applied to the French setting. We adjusted the values of the implemented fitness cost of drug resistance for *S. pneumoniae* associated with lower transmissibility so that the model simulations do not exceed 20% of resistance at baseline.

Details, I would not use the word rate for the proportion/fraction resistance. For me, the rate is an incidence per time unit.

We changed this throughout the manuscript and removed the term rate.

In line 263, I think mentioning of a relative increase may be more intuitively understandable than an absolute increase.

We made the appropriate change.

Reviewer #2 (Recommendations for the authors):In this manuscript the authors propose a mathematical model to explore the effects of the COVID-19 pandemic on the epidemiology of S. pneumoniae, considering both direct effects of SARS-CoV-2 infection, which is assumed to be capable of increasing the rate of progression from colonization with S. pneumoniae to invasive diseases, and indirect effects. The indirect effects are assumed to be the result of (i) changes in contact patterns between individuals (eg through population-wide "lockdowns" and isolation of COVID-19 cases), and (ii) changes in antibiotic consumption which is assumed to be the result of both widespread antibiotic prophylaxis amongst those infected with SARS-CoV-2 and changes in healthcare seeking behaviour. The questions addressed are of widespread interest, and the authors are to be congratulated for a first attempt at addressing this difficult problem. Strengths of the work include a well-motivated research question, a largely clear exposition of the model and findings, and a generally well-written manuscript. I have no concerns about the implementation of the models. I do, however, have a number of concerns about the model assumptions that lead me to question whether the observed changes in the incidence of invasive disease with S. pneumoniae and in the prevalence of resistance amongst invasive isolates are really driven by the mechanisms encapsulated by the proposed models. I expand on these below.

We express our gratitude for acknowledging and appreciating our efforts. We also thank the reviewer for raising these different points that were addressed in this new version. We believe that we now encompass most of possible mechanistic hypotheses that could affect pneumococcal epidemiology trends over the study period.

One concern is that this study focuses on the community transmission of bacteria, and (unless I have fundamentally misunderstood the work) a central assumption seems to be that high rates of antibiotics were used in the community as prophylaxis amongst patients with COVID-19. As S2.2 explains, in the scenarios considered 18.1%, 33.0%, and 55.1% of patients received antibiotic prophylaxis during each COVID infection. The authors state "As reported in Langford et al. [3], a meta-analysis on antibiotic prescribing in patients with COVID-19, prophylactic antibiotic prescribing in the inpatient/outpatient setting was estimated at 59.3%. Therefore, the values above that we use in our model are conservative compared to this estimate." There are two issues here: first I couldn't find where the number of 59.3% in the Langford et al. paper came from (searching for 59 in the manuscript gave no relevant hits). In fact, this paper reports even higher rates of antibiotic use "The majority of patients with COVID-19 received antibiotics (71.9%, 95%CI 56.1 to 87.7%)." (elsewhere in the paper also reported as 71.8%). The second (and more important) issue is that even though the scope of Langford et al. included hospital and community settings, in all 24 of the included studies the setting was either ICU (n=5) or hospital (n=19). There is no reason to think these numbers are at all representative of the community setting (the focus of the manuscript) and the assumption that in 18% of COVID-19 infections (including, presumably, subclinical infections) antibiotics were taken prophylactically in the community does not seem credible to me (indeed, from personal experience, I am not aware of a single person who took antibiotics prophylactically in the community while infected with COVID). This might well just reflect my ignorance – perhaps there were countries where antibiotic prophylaxis in the community was widespread, but this has not been shown in the paper by Langford et al., and given that this is a central premise of the model and something that is presumably driving a lot of the results this is surely something where the data supporting this assumption needs to be made very clear if indeed such data exists. If, as I suspect is true in most settings, antibiotic prophylaxis during COVID infection in the community was negligible then I do think there is a need to make appropriate changes to the manuscript.

First of all, we believe that Reviewer #2 was referring to a different paper by the same author. We drew our antibiotic prescribing estimates listed in the appendix from:

Langford BJ, So M, Raybardhan S, Leung V, Soucy J-PR, Westwood D, et al. Antibiotic prescribing in patients with COVID-19: rapid review and meta-analysis. Clin Microbiol Infect. 2021;27: 520–531. (https://doi.org/10.1016/j.cmi.2020.12.018).

while we believe that the Reviewer #2 is referring to:

Langford BJ, So M, Raybardhan S, Leung V, Westwood D, MacFadden DR, et al. Bacterial coinfection and secondary infection in patients with COVID-19: a living rapid review and metaanalysis. Clin Microbiol Infect. 2020;26: 1622–1629. doi:10.1016/j.cmi.2020.07.016 (https://www.sciencedirect.com/science/article/pii/S1198743X20304237?via%3Dihub)

Reference to 59.3% can be found on the bottom of page 523 of the Langford et al., 2021 manuscript (https://doi.org/10.1016/j.cmi.2020.12.018), direct citation below:

“Antibiotic prescribing by setting/severity

Antibiotic prescribing was lowest in the mixed inpatient/outpatient setting at 59.3% (95% CI 38.7% to 77.1%), followed by the inpatient hospital setting at 74.8% (95% CI 67.8-80.7%) and highest in the ICU setting at 86.4% (95% CI 73.7-93.6%).”

In Langford et al., 2021, out of 154 studies in total included in the meta-analysis, 12 studies included mixed hospital/outpatient setting that we were interested in (pg. 523, Table 1). These 12 publications from mixed hospital/outpatient setting are also listed out in Langford et al., 2021 supplement. 4062 patients in total from mixed hospital/outpatient setting were included in the analysis and it was estimated that 59.3% used antibiotics (pg. 525, Figure 5).

This was the largest systematic review of antibiotic use that included COVID cases in outpatient setting and we agree that mixed hospital/outpatient setting estimates of antibiotic use (59.3%) might not be the ideal representative of antibiotic use for COVID cases in the community setting.

For this reason, in the previous version of our article, we assumed that antibiotic use in COVID19 infected individuals in the community will be lower than this estimated number for a mixed setting, and we tested three possibilities in our scenarios using lower values (18.1%, 33.0%, and 55.1%). For example, Tsay et al., 2022 JAMA (doi:10.1001/jama.2022.5471) found that in the US population (age 65+) during the first year of the COVID-19 pandemic, 30% of outpatient visits for COVID-19 among Medicare beneficiaries were linked to an antibiotic prescription, 50.7% of which were for azithromycin. This suggests that the use of antibiotics for treating COVID infection differs between age groups (and possibly countries), and, in our model, we wanted to test a wide range of antibiotic exposures.

Interestingly, reported trends in antibiotic consumption over the period highlight an increase in macrolide consumption (Nandi et al. 2023 e Clinical Medicine). Following discussions with experts, we also investigated the impact of incorporating specifically addressed azithromycin exposure (by comparison to β-lactams) during the early stages of the pandemic (Appendix 2 – Figure 2) to account for differences in resistance selection due to longer drug persistence in the body.

In this new version, we investigated lower levels of antibiotic use in COVID-19 patients by varying azithromycin exposure from 0% to 20% of COVID-19 infected individuals. This is outlined in detail all model assumptions in the new version of the manuscript.

A second concern is in the treatment of the progression from colonisation to invasive disease. The authors assume that infection with SARS-CoV-2 increases the rate of progression (and consider up to 100-fold increases). This itself is reasonable. However, as the authors acknowledge, other respiratory viral infections also have this effect (SI-8: "estimations show that influenza co-infection increases the rate of progression from S. pneumonia colonization to invasive pneumococcal disease between 49 and 146-fold, on average, depending on the age group"). Since a major effect of the changes in contact patterns in response to the pandemic was that in many countries other respiratory infections, including influenza, did not circulate as usual, wouldn't this be expected to also have a substantial impact on the rate of progression from colonization to invasive disease? i.e. if it is important to consider the impact of SARS-CoV-2 on the progression, why not also consider the effect of other viral pathogens at the same time? A similar concern applies to antibiotic use. We know that in interpandemic periods a lot of antibiotic use in the community is given in response to viral respiratory infections, and since we know that changes in contact patterns had a profound impact on many such viral respiratory infections, is it not also likely that these had an effect on antibiotic use? Or is the assumption that such changes are just absorbed into changes in health-seeking behaviour?

We agree that it is very important to acknowledge the impact of reduced influenza and other respiratory virus circulation on the rate of progression from colonization to invasive disease for *S. pneumoniae* and we thank you for this suggestion, it is something we did not incorporate in the prior version of the model. It is well-known that viral infections can create a conducive environment for invasive pneumococcal diseases. Therefore, the reduction in influenza (and other respiratory viruses’) transmission over the pandemic period and more specifically during the lockdowns likely contributed to the decrease in IPDs.

To verify this hypothesis, we introduced an additional parameter into our model that modulates the progression from colonization to disease. Specifically, we considered a reduced invasion risk during the lockdown period and other periods throughout the year when influenza and other ILIs were circulating less than usual. According to recent publications describing carriage studies during the lockdowns in different countries, we also included a new constraint regarding pneumococcal carriage during the lockdown. We indeed assumed that the levels of bacterial colonization remained constant or decreased only slightly during the lockdown period due to reduced social contacts. This new insight into pneumococcal epidemiology over that period gives an advantage to the hypothesis of a reduction of infection risk associated with reduced viral transmission rather than a strong change in transmission and carriage due to lockdown and contact restrictions.

Changes in the community antibiotic use for any reason are absorbed into changes in the healthcare-seeking behavior during the pandemic. Across 31 scenarios, we assume a decrease of 18% in the overall community antibiotic use based on data from France in 2020 (Bara et al., 2022 *Antibiotics*, doi: 10.3390/antibiotics11050643).

A third concern is really a question about whether there might be another mechanism that could be explaining the generally increasing trends in penicillin and macrolide resistance in invasive pneumococcal disease summarised in figure 1. We know that the S. pneumoniae serotypes which have a longer carriage duration are more likely to be resistant (see, for example, Lehtinen et al., PNAS 2017), and also that these have a higher reproduction number (the shorter duration of carriage serotypes being maintained by negative frequency-dependent selection). We would therefore surely expect these less resistant serotypes (with lower reproduction numbers and shorter durations of carriage) to be more impacted by social distancing measures as they would be the first serotypes to have reproduction numbers reduced to below one. This would then be a completely different mechanism to explain observed changes in resistance patterns. Clearly, a lot of work would be needed to establish whether or not this can explain observed changes, though I feel the same is true of the explanatory mechanisms proposed in this paper (increased antibiotic use) given that the current paper lacks any country-specific antibiotic use data or attempt to see if different changes in patterns of antibiotic use can explain the different changes in resistance seen in different countries.

The duration of bacterial carriage, as examined by Lehtinen et al. PNAS 2017, is a hypothesis that warrants further exploration in the context of the COVID pandemic. We tested this hypothesis in our model as an additional mechanism and described it in the new version of the manuscript. Current manuscript describes country-specific antibiotic use (France) and the model tests different changes in patterns of antibiotic use that reflect different age groups. After assessment, the assumption on duration could not explain on its own the changes during 2020.

We believe that most obvious and plausible mechanisms are now included and assessed in this new version. They were confronted to experts of the domain.

1. lines 118-120 highlight azithromycin as a particular concern for overuse, but this is difficult to square with the review the authors cite from Langford which reports macrolides as accounting for only 6.5% of antibiotic prescribing in COVID patients. This review suggests fluoroquinolones and β-lactams accounted for over 80% of prescribed antibiotics

In the updated version of the manuscript, we have incorporated numerous additional studies documenting a surge in azithromycin utilization during the initial stages of the pandemic.

Similar to our response to the first comment, we were referring to Langford et al., 2021 article: Macrolides accounted for 18.9% of antibiotic prescribing in COVID patients. On the other hand, an article doi:10.1016/j.cmi.2020.07.016, also by Langford focuses on the increased use of antibiotics in the incidence of bacterial co-infections presenting with COVID, and provides evidence to support studies by Knight *eLife*, 2021 and Rusic in Life, 2021, expressing concern that there is COVID-19 associated antimicrobial overuse or misuse. Study by Rusic, 2021 is the one that specifically refers to azithromycin use during the early pandemic.

Citation below from Langford et al., 2021 (paper not cited on line 121, but the one we can add here as well states): “Of ten classes of antibiotics studied, the most common antibiotic classes prescribed were fluoroquinolones (n = 612, 20.0%), macrolides (n = 579, 18.9%), b-lactam/blactamase inhibitors (n = 459, 15.0%) and cephalosporins (n = 459, 15.0%).”

During the first lockdown in France, community azithromycin consumption increased by 25.9%, with the increase varying from 13.4% to 47.3% depending on the week (Weill et al., 2021), while the overall number of azithromycin prescriptions across France in 2020 increased by 10.1% relative to 2019 (Bara et al., 2022).

2. On first looking at the manuscript I was quite concerned that the changes in resistance in figure 1 might reflect changes in health-seeking behaviour rather than real changes in the underlying epidemiology, for example, if certain types of isolates are less likely to be resistant and these types were reported less frequently. Now that I realise that the data in figure 1 all come from invasive isolates (as stated on line 176) this explanation seems less likely. It would be helpful if figure 1 could be labelled appropriately to make it clear that the data shown represents only invasive isolates.

We thank the reviewer for raising this was unclear. We now changed this to “invasive isolates” both within text and in the figure caption.

3. line 248 "some respiratory viruses are known to favor bacterial disease" I don't think "favor" is the right word here. Perhaps "increase the risks of"?

We made the appropriate change.

4. Figure 3 would be easier to read if the subgraphs were labelled with the scenario. Also, it wasn't clear to me what the three graphs in panel C represent.

Figure 3 has now been replaced with Table 1.

5. Table S3. I think A=20 in all the scenarios so that a minimum of 18% of patients infected with SARS-CoV-2 are given antibiotics prophylactically. It would increase readability if this was made explicit in the table.

Parameter A has now been removed from the model and replaced with separate compartments for azithromycin exposure.